# Recent Advances in Synthesis, Modification, Characterization, and Applications of Carbon Dots

**DOI:** 10.3390/polym14112153

**Published:** 2022-05-25

**Authors:** Arul Pundi, Chi-Jung Chang

**Affiliations:** Department of Chemical Engineering, Feng Chia University, 100, Wenhwa Road, Seatwen, Taichung 40724, Taiwan; arul.arjun585@gmail.com

**Keywords:** carbon dot, doped, kilogram-scale fabrication, photoluminescence, electron transfer, NEXAFS, sensor, photocatalyst, bioimaging

## Abstract

Although there is significant progress in the research of carbon dots (CDs), some challenges such as difficulty in large-scale synthesis, complicated purification, low quantum yield, ambiguity in structure-property correlation, electronic structures, and photophysics are still major obstacles that hinder the commercial use of CDs. Recent advances in synthesis, modification, characterization, and applications of CDs are summarized in this review. We illustrate some examples to correlate process parameters, structures, compositions, properties, and performances of CDs-based materials. The advances in the synthesis approach, purification methods, and modification/doping methods for the synthesis of CDs are also presented. Moreover, some examples of the kilogram-scale fabrication of CDs are given. The properties and performance of CDs can be tuned by some synthesis parameters, such as the incubation time and precursor ratio, the laser pulse width, and the average molar mass of the polymeric precursor. Surface passivation also has a significant influence on the particle sizes of CDs. Moreover, some factors affect the properties and performance of CDs, such as the polarity-sensitive fluorescence effect and concentration-dependent multicolor luminescence, together with the size and surface states of CDs. The synchrotron near-edge X-ray absorption fine structure (NEXAFS) test has been proved to be a useful tool to explore the correlation among structural features, photophysics, and emission performance of CDs. Recent advances of CDs in bioimaging, sensing, therapy, energy, fertilizer, separation, security authentication, food packing, flame retardant, and co-catalyst for environmental remediation applications were reviewed in this article. Furthermore, the roles of CDs, doped CDs, and their composites in these applications were also demonstrated.

## 1. Introduction

CD is a zero-dimensional (0D) quasi-spherical carbon nanomaterial with a size ranging from 1 to 10 nanometers. Compared with traditional quantum dots, such as metal or semiconductor-based QDs, carbon-based QDs (such as carbon QDs, graphene QDs, and polymer QDs) have excellent biocompatibility and low cytotoxicity. CDs have some unique properties, such as facile synthesis, ease of functionalization, excellent water dispersibility, excellent light-harvesting, sharp fluorescent emission spectrum, up-conversion, tunable photoluminescence properties, effective electron transfer, and good photostability. Therefore, CDs have been applied to a wide range of applications, including bioimaging, biomedical and chemical sensing, therapy, optoelectronics, solar cells, fertilizer, separation, security authentication, food packing, and co-catalyst for environmental remediation applications. The continuous efforts devoted to CDs-related research have significantly explored some important knowledge about CDs-based materials and improved their performance in different fields. However, some serious challenges, such as inadequate and complicated purification, difficulty in large-scale production, low quantum yield, and ambiguity in understanding the structure-property correlation, are still the major obstacles to the commercial use of CDs in various applications.

In 2022, some review articles have provided useful information about the research of CDs. Ðorđević et al. [1] reported the pre-synthetic and post-synthetic approaches for CDs synthesis and the influences of chemical tools on the properties of CDs. Wu et al. [2] reviewed the synthesis of four types of CDs-based composites (metal-CDs, organic-CDs, nonmetallic inorganic-CDs, and multicomponent-CDs composites) and their uses in the bioapplications, such as biosensing, bioimaging, and drug delivery. Zhang et al. [3] reviewed the development of synthetic/modification methods of CDs and CDs-based optical sensors for pesticides, as well as the advantages, possible limits, and challenges for their real-world application. Saini et al. [4] summarized the new findings related to the use of CDs, doped-CDs, and their composites for visible-light-driven photocatalysis. Wang et al. [5] reported the utilization of CDs and CDs-based composites for the detection and adsorption of radioactive ions that may cause long-term influence on the environment and hurt the human body by entering the food chain. Zhai et al. [6] provided a review of the latest study related to the use of CDs for energy storage and electrochemical processes’ applications, including CO_2_ reduction, H_2_/O_2_ production, supercapacitors, and batteries. Han et al. [7] summarized the advances in CDs-based ratiometric fluorescent sensors for food safety applications and detection mechanisms. Korah et al. [8] reviewed three aspects of CD-drug interactions, including drug detection, photocatalytic degradation by CDs-based composites, and enhancement of drugs by using CDs. Mohammadi [9] reported the recent progress in using carbon-based QDs as a biosensing and fluorescence imaging platform for the early stage diagnosis of cancers. However, there is still an urgent need for the exploration of some important issues, such as the roles of CDs in different applications, kilogram-scale fabrication of CDs, and doping/surface modification, as well as the correlation among process parameters, structures, compositions, properties, and performances of CDs-based materials.

This review mainly focuses on the preparation (synthesis, modification, and purification), the key factors affecting the properties/performance, and possible applications of CDs, as well as the roles of CDs in these applications. We start with the advances in synthesis approach, purification methods, kilogram-scale fabrication, and modification/doping methods. We illustrate some examples to correlate process parameters, structures, compositions, properties, and performances of CDs-based materials. The synchrotron NEXAFS test was utilized for analyzing the electronic structures and photophysics of CDs-based materials. The properties and performance of CDs can be tuned by some synthesis parameters, such as the incubation time and precursor ratio of the microwave-assisted process, the laser pulse width of the laser ablation synthesis, and the average molar mass of polymeric precursor. Surface passivation also has a significant influence on the particle sizes of CDs. Moreover, some factors affect the properties and performance of CDs, such as the polarity-sensitive fluorescence effect and concentration-dependent multicolor luminescence, together with the size and surface states of CDs. Recent advances of CDs in bioimaging, sensing, therapy, energy, fertilizer, separation, security authentication, food packing, flame retardant, and co-catalyst for environmental remediation applications are reviewed in this article. Furthermore, the roles of CDs, doped CDs, and their composites in bioimaging, catalysis, energy, sensing, therapy, agriculture, food packing, flame retardant, anti-counterfeit, and separation applications are also summarized.

## 2. Preparation and Modification of CDs

### 2.1. Synthesis Methods

CDs can be synthesized via breaking bulk carbon structures (top-down process) or via the chemical reactions of precursors (bottom-up method). Various CDs and doped CDs were prepared and used in various applications. Their optical, electrical, and surface chemical properties can be tuned by changing the precursors and reaction parameters. 

#### 2.1.1. Top-Down Approach

Various physical and chemical methods were applied to break the bulk carbon structures for the preparation of CDs by using the top-down methods. CDs with high crystallinity and intact structures can be obtained by the top-down methods, especially the graphene quantum dots (GQDs) with typical graphitic structures. 

##### Laser Ablation

Laser ablation is a facile and rapid process for the synthesis of CDs in which a laser irradiates carbon targets to remove material from the solid surface. Li et al. [10] fabricated twin graphene quantum dots (GQDs) by the electric-field-assisted temporally shaped femtosecond laser ablation liquid of the graphene dispersion. GQDs with similar size but different oxygen contents and crystallinity were obtained via adjusting the electric field intensity and the pulse delay of the femtosecond laser. Shen et al. [11] synthesized GQDs by the femtosecond laser ablation process in liquid. The laser-induced graphene is the carbon source. Moreover, by incorporating ammonia water into the graphene dispersion, N-doped graphene quantum dots can be prepared for the Fe^3+^ sensing application. Zhang et al. [12] studied the synthesis of carbon quantum dots by using the pulsed laser ablation method, which can effectively inhibit Aβ42 peptide aggregation and reduce the cytotoxicity of the Aβ42 peptides. The graphite target pellet was irradiated by a pulsed Nd:YAG laser to ablate the pellet. Kaczmarek et al. [13] synthesized fluorescent carbon dots (CDs) by a nanosecond laser ablation of a graphite target immersed in polyethyleneimine and ethylenediamine and the separation by a dialysis process. Most of the molecular fluorophores are associated with carbogenic nanostructures for the ablation of polyethyleneimine, while, in the case of ethylenediamine, free fluorescent molecules are the dominant products. The ablated particles (including atoms, ions, and atom clusters) were heated to a high temperature, and they interacted and reacted with the surrounding liquid molecules in the cavitation bubbles. Then nanoparticles formed during the plasma-cooling period.

##### Electrochemical Method

Zhou et al. [14] reported the mass production (9.36 g per batch) of N-doped carbon quantum dots (N-CQDs) by an electrochemical setup, using prebaked carbon as the anode and ammonium bicarbonate as the electrolyte. The process with an efficient utilization rate of raw material (90.18 wt.%) and a high conversion (1.87 g_(N-GQDs)_g_(PCA)_^−1^) is promising and practical for the fabrication of N-CQDs. Zhou et al. [15] reported an electrochemical exfoliation process for the synthesis of GQDs. The solid-liquid interfacial exfoliation technique was carried out in a two-electrode cell. Carbon fiber bundles, Ti mesh, and H_2_SO_4_ were used as the anode, cathode, and electrolyte, respectively. Danial et al. [16] reported the production of a suspension of graphene quantum dots (GQDs) by using an electrochemical exfoliation process that involved an electrolyte solution containing citric acid and NaOH, and pristine graphite rods-based electrode.

#### 2.1.2. Bottom-Up Approach

CDs prepared by bottom-up methods have amorphous carbon cores, many surface functional groups, and doping sites. Many researchers synthesized the QDs by the hydrothermal/solvothermal approach. The thermal decomposition of precursors in aqueous or organic solution at temperatures typically between 100 and 250 °C. CDs can also be prepared by microwave-assisted hydrothermal/solvothermal processes. Compared with traditional hydrothermal/solvothermal methods, the microwave approaches can effectively reduce the reaction time for the CDs synthesis [17]. The crystallinity, size, and solubility of the CDs can be tuned by changing the power, reaction temperature, reaction time, and types of precursors. Precursors can be citric acid, saccharides, ethylene glycol, and even the abundantly available biomass, such as waste *cotton* linter [18], *olive* solid wastes [19], *orange* juice [20], *bitter apple* peel [21], *shrimp* egg [22], *muskmelon* peel [23], *ginkgo* leaves [24], eggshell membranes [25], extract of waste tea [26], *egg* yolk [27], *pine* pollen [28], *Colocasia esculenta* leaves [29], *Camellia japonica* flowers [30], and other organic wastes [31,32].

##### Hydrothermal, Solvothermal Method

Simple, doped, and supported carbon dots can be fabricated by the facile and low-cost hydrothermal or solvothermal method [33]. The carbon dots prepared with a hydrothermal approach, using polyacrylamide and sodium citrate precursors, are water-soluble and exhibit strong fluorescence [34]. The bright-blue-fluorescent CDs showed an “off-on” sensing response with the addition of Pb^2+^ and pyrophosphate. The probe was found to be validated with the sensing of human urine and real water samples. N-doped CDs (N-CDs) were synthesized by a hydrothermal process, using cellobiose, the hydrolysis product of cellulose, and the dimer of glucose [35]. Cellobiose and NH_4_HCO_3_ acted as the C and N sources, respectively. N-CDs can be used for phosalone detection and temperature monitoring ranging between 10 and 80 °C. Newman et al. [36] reported the preparation of ethylenediamine (EDA)-doped and L-phenylalanine (L-Phe)-doped CDs from the natural palm oil waste, palm kernel shells, via the hydrothermal and solvothermal methods. These CDs show excellent photoluminescence properties and the quantum yield of 13.7% and 8.6% for EDA-doped and L-Phe-doped N-CDs, respectively. It confirms the feasibility of producing high-quality CDs by using palm-kernel-shell biomass as the precursor. 

##### Microwave-Assisted Synthesis

The microwave-assisted process is faster than traditional solvothermal and hydrothermal methods for the synthesis of nanomaterials [37]. CDs can be synthesized by the microwave-assisted process in which electromagnetic radiation is applied to the precursors. Laddha et al. [38] prepared CDs by a microwave-induced decomposition of glutathione, polyethylene glycol (PEG-400), and H_3_PO_4_ precursors. Highly fluorescent (quantum yield 15%), water-soluble, and blue-emitting N-S-P co-doped CDs were prepared and used for selective and sensitive sensing of toxic Cr(VI) ions. Uriarte et al. [39] fabricated carbon dots from glycerol and urea precursors by a microwave-assisted process. Since tetracycline can quench the fluorescence of CDs, these CDs can act as probes for tetracycline sensing in urine samples with high recoveries and precision. Blue-fluorescent CDs were prepared from the palm kernel shell (PKS) biomass by the microwave-assisted approach through a four-stage mechanism, namely the dehydration of PKS, polymerization, nucleation, and growth [40]. The CDs have the potential to act as fluorescent inks and cell imaging probes for both Gram-positive and Gram-negative bacteria. Various biomass such as orange peels and prickly pear can be used to prepare carbon quantum dots or doped carbon dots for Cr (vi) sensing or photosensitizer-related applications [41,42]. Other CDs synthesized by microwave-assisted synthesis can be used as fluorescent ink, logic gate operation, and probes to detect various analytes such as ammonia, Cr (VI), ascorbic acid, copper, and etidronate disodium [43,44,45].

##### Ultrasonic Methods

The sonochemical process is one of the well-developed methods for the CDs synthesis because it is a low-temperature, rapid, facile, non-toxic, and less expensive process [46]. Xu et al. [47] reported the synthesis of N-doped M-CDs with green, chartreuse, and pink emitting properties by ultrasonic treatment of the kiwifruit juice, using different additive reagents, including ethylenediamine, ethanol, and acetone. These CDs are useful fluorescent inks that can be applied in the anticounterfeit field, monitoring trace species, and applications in food chemistry. Graphene quantum dots (GQDs) prepared by the ultrasonic process were incorporated on the surface of amine-functionalized Si nanoparticles (SiNPs) through the amide linkage to make the GQDs-SiNPs nanocomposite [48]. The GQDs-SiNPs/GC electrode was proved to have high activity for glutathione sensing.

#### 2.1.3. Fabrication of Kilogram-Scale CDs

CDs have attracted a lot of attention due to their useful merits. However, the complicated synthesis and purification processes limit their large-scale synthesis and wide application. After a lot of effort devoted to the research, there are recent breakthroughs in developing scale-up CD synthesis processes. An aldol condensation method at ambient temperature and pressure was proposed by Li et al. [49]. It is a low-cost and effective method for the large-scale synthesis of CDs, which can produce 1.083 kg CDs in 2 h. Meanwhile, CDs can be functionalized by doping with nitrogen or sulfur/nitrogen. Ji and co-workers [50] reported the fabrication of kilogram-scale CDs from o-phenylenediamine with a high yield (over 96%) by the facile purification process at a low cost (0.1 dollar/g). Figure 1 shows the synthesis process of C-dots_606_ (CDs with emission at 606 nm) from *o*-PDA at a kilogram scale (1.104 kg) with high yield by a combined microwave-hydrothermal process. The optical properties of CDs can be adjusted via protonation-deprotonation. A red shift of 47 nm for the fluorescent emission wavelength of CDs can be achieved by protonation. The photoluminescent QE of CDs can be enhanced by three times via deprotonation. Li et al. [51] reported a convenient green kilogram-scale preparation of fluorescent CQDs from poplar leaves through a hydrothermal method. The throughput of CQDs reaches a high level of 1.4975 kg per pot. A cost-efficient and robust approach for the large-scale preparation of CDs was developed by Fang et al. [52] through microwave-assisted carbonization of inexpensive industrial surfactants. Four surfactants were selected as precursors to fabricate CDs on a kilogram scale, which also presented no limitation for further enlargement.

### 2.2. Purification of Carbon Dots

The products obtained from the reaction usually contain the mixtures of some CD fractions, molecular intermediates, and several side products. It was reported that many impurities might be formed when CDs were synthesized from waste materials or biomass. The byproducts formed during the synthesis of CDs may change their fluorescent emission properties [53]. Thus, the separation and purification of these products are critical for fabricating CDs nanomaterials with acceptable purity and consistency [54]. CDs can be purified by several processes, including centrifugation, dialysis, filtration, electrophoresis, column chromatography, and high-performance liquid chromatography [55,56,57]. Centrifugation and syringe filtration cannot effectively remove the unreacted precursors and reaction by-products. Dialysis is a widely used method for the purification of CDs by removing unreacted products and nanoparticles that are smaller than the expected CDs [58,59]. Selecting the dialysis membrane with the appropriate pore size is important to achieve pure CDs and prevent product loss. Chen et al. [60] reported that there was no standard for the dialysis time or the selection of the molecular weight cutoff (MWCO) of the dialysis membrane. González-González et al. [61] performed the purification of CDs via the ultrafiltration and dialysis processes by using the Spectra-Por^®^ Float-A-Lyzer^®^ G2 with a membrane of 3.5–5 kDa. A Dionex ICS-1600 Ion Chromatography System was used after the dialysis process to ensure the complete purification of CDs. Jia et al. [62] used the dialysis bags (14000 Dalton cutoff) to remove contaminations and unreacted species.

### 2.3. Doping of CDs

Since doping can effectively tune the physical and chemical properties of CDs, it has recently attracted increasing attention. Doping is the introduction of a dopant, such as boron, nitrogen, chlorine, sodium, and potassium, into the structure of a CQD. Doped CDs and CDs-based composites exhibit enhanced light absorption and photoluminescence properties compared to pristine CDs. When CDs were doped with appropriate heteroatoms, their chemical composition, nanostructure, electronic structure, and catalytic properties changed because of the overlapped atomic orbitals of carbon atoms and heteroatoms, together with the electron push-pull effect of heteroatoms [63]. Metal ions-doped CDs may exhibit more noticeable changes in optical, electronic, and magnetic properties than non-metallic atom-doped CDs because metal ions have a larger atomic radius, more electrons, and unoccupied orbitals. Cu-CDs were prepared with Cu(Ac)_2_ [64], Na_2_[Cu(EDTA)] [65], CuCl_2_ [66], Cu(NO_3_)_2_ [67], and CuSO_4_ [68] as precursors, respectively. Iron-doped CDs were synthesized by introducing iron benzoate [69], FeCl_3_ [70], or Fe-gluconate [71] as precursors. Moreover, other metal-doped CQDs have also been synthesized and applied for different applications, such as manganese-doped CDs [72], Zn-doped CDs [73], cobalt(II)-doped CDs [74], gadolinium-doped CDs [75], europium-doped CDs [76], Ce doped-CDs [77], terbium-doped CDs [78], europium-doped CDs [79], ruthenium-doped CDs [80], and germanium-doped CDs [81].

For the non-metallic atom-doped CDs, elements close to C in the periodic table, such as B, N, S, and P, can be incorporated as dopants into the carbon frameworks [82]. N-doped and S-doped CDs are widely studied among the reported non-metallic atom-doped CDs. N-doping is commonly used for the synthesis of doped CDs because of the comparability of the valence electrons and radius of the N atom, allowing effective tuning of optical and electronic properties of CDs [83]. The S atom has similar electronegativity but has a larger atomic size than the C atom. Hence, the S atom exhibits easier electronic transition than that of the C atom, allowing S doping to effectively improve the properties of CDs [84,85]. The doped CDs can be used as fluorescent probes for the sensing of various analytes [86]. The doped CDs were also applied to other applications, such as fluorescence imaging [87,88] and magnetic resonance imaging [89].

### 2.4. Surface Modification

The properties of CDs were affected by the types of surface functional groups. The surface-modified CDs can be used for light-emitting devices, drug releasing, chemical sensing, targeting, and extracting analytes [90]. The formation of surface functional groups such as hydroxyl, carboxyl, amine, and amide can impart higher dispersing stability of CDs in many aqueous media and solvents. It helps to improve the performance catalysis, sensing, and bio-related applications of CDs-based materials by facilitating their interaction with pollutants, analytes, and biological species. Moreover, the characteristics of graphene quantum dots can be tuned by functionalizing their edge structures. The surface modification of CDs can be achieved by (1) the intermolecular interactions, namely complexation, chelation, electrostatic interactions, and loading polymers (such as propionylethylenimine-co-ethylenimine [91] and polyethylene-glycol [92]); and (2) the covalent modification of carboxyl, amino, and hydroxy groups on CDs. Tachi et al. [93] successfully elevated the quantum yield (QE) of graphene quantum dots (GQDs) by restricting the vibration and rotation of the functional groups at the edges of GQDs through an esterification reaction with benzyl alcohol. The graphene-stacked structures participated in the π-π interaction with adjacent aromatic rings of the benzylic ester, impeding the nonradiative recombination process in GQDs. Meierhofer et al. [94] reported the synthesis of two types of CDs by the reaction between citric acid and diaminopyridine (DAP) or polyethylenimine (PEI). The CDs consisted of hybridized carbon cores and surface functional groups (Figure 2). The chemical nature of the functional groups (such as hydroxyl, carboxyl, amino, epoxy, and amides) depends on the types of precursors, reaction conditions (including solvent, reactant ratio, time, temperature, pH, etc.), and purification processes (including centrifugation, column chromatography, and dialysis). The selected precursor decides the types and amounts of surface groups and fluorophores of CDs, changing their optical and pH-responsive properties.

## 3. Key Factors

### 3.1. Synthesis Parameters Affecting the Properties/Performance of CDs

#### 3.1.1. Process Parameters

(1)Microwave-assisted process: incubation time and precursor ratio:

Xu et al. [95] synthesized the nitrogen and sulfur (N,S-CDs)-co-doped CDs via a microwave-assisted process, using cystine as a source for C, N, and S. CDs with high solubility in an aqueous solution can be applied for the turn-on fluorescence Hg(II) sensing in lake water and spiked tap samples. The precursor ratio and incubation time have been optimized to achieve high fluorescence of CDs.

(2)Laser ablation synthesis: laser-pulse width

Hu et al. [96] investigated the influence of laser-pulse width on the morphology and size of CDs prepared by a laser ablation method. The results revealed that the nucleation and growth of synthesized particles were tuned by changing the pulse width of the laser. After being refluxed in HNO_3_ and the surface passivation with organic molecules (PEG1500N or poly(propionylethyleneimine-co-ethyleneimine (PPEI-EI)) [97], the obtained CDs exhibit bright emitting properties. Compared with a short-pulse-width laser, using a long-pulse-width laser for the laser ablation process exhibited better control of the size and morphology of CDs.

#### 3.1.2. Average Molar Mass of Polymeric Precursor

Jiang et al. [98] synthesized a series of CDs from polyethylene glycol (PEG) precursors with various molecular weights. The resulting CDs showed low cytotoxicity and amazing performance for cellular bioimaging. They found that CDs were formed from reactions of PEG via the “oxidized decomposition-crosslinking-carbonization” process (Figure 3a). The emission wavelengths and surface functional groups of CDs samples prepared by various PEGs were similar. However, the molecular weight of the PEG precursors has a great influence on the fluorescence intensities, thermal stability, particle sizes, and element ratio of CDs. It provides a useful and convenient way for tuning the properties of CDs. These CDs exhibited low cytotoxicity and excellent cell-imaging performance (Figure 3b).

#### 3.1.3. Surface Passivation Effect

Kang et al. [99] reported the synthesis of amino-functionalized GQDs (FGQDs) for Fe^3+^ sensing application, using graphite and polypyrrole (PPy) as the carbon and nitrogen precursors, respectively. Compared with pristine GQDs, FGQDs exhibited a narrow size distribution and small particle sizes due to the “surface passivation effect” of the polypyrrole surfactant. When the pulse laser is exposed to the target (such as graphite flakes) during the pulsed laser ablation process, the partial decomposition of the graphite flakes, ethanol, and PPy. Ref. [100] lead to the formation of C, H, O, and N precursors. Then the FGQDs formed due to the high surface energy of precursors. The aggregation of FGQDs can be effectively suppressed via the loading of PPy because PPy and FGQDs have a positive charge and negative polarity, respectively [101]. The formation of PPy passivation layers around FGQDs leads to the smaller size and a narrower particle size distribution of FGQDs (surface passivation effect).

#### 3.1.4. Purification by Preparative Column Chromatography

CDs with a distribution of structures may be obtained because different reaction pathways between the precursors are possible during the hydrothermal, solvothermal, or microwave synthesis process. Usually, the synthesized CDs dispersion is purified by filtration and dialysis processes. Essner et al. [102] found that the CDs were not properly dialyzed in a lot of CDs-related papers. Their results revealed that the ion-sensing properties of CDs are much better than that of the side products, indicating that the CDs should be adequately purified to improve their performance. 

Furthermore, it takes a longer time to purify the CDs during the dialysis process. Hinterberger et al. [103] reported the purification and separation of CDs solutions into various fluorescent species (free fluorophores, fluorophores bound on CDs, and low-fluorescent carbon particles without fluorophores) by the preparative column chromatography. The fabrication of purified CDs by the chromatography process exhibits some advantages, such as discovering the exact internal structure of CDs, improving CD properties for industrial applications, and removing toxic precursor/side-products to enhance the biocompatibility of CDs. In addition, Michaud et al. [104] reported the complete separation of N-CDs and other components by a two-step gradient chromatography method, including column chromatography and high-performance liquid chromatography processes.

### 3.2. Factors Affecting the Properties/Performance of CDs

The strong and tunable fluorescence of CDs has been widely investigated because fluorescence enables the use of CDs in biomedicine, optics, catalysis, and sensing applications. Factors affecting the properties or performance of CDs are worth studying, including the size-dependent fluorescence, concentration-dependent multicolor luminescence, solvation effects, electronic structures, and photophysics analysis. 

#### 3.2.1. Concentration-Dependent Multicolor Luminescence

The development of color-tunable fluorescent materials is quite important for light-emitting diode applications. He et al. [105] fabricated CDs by the hydrothermal method, using humic acid precursor. The CDs exhibited concentration-dependent multicolor luminescence in deionized water, N,N-dimethylformamide, and formic acid upon the UV light (λ = 365 nm) excitation (Figure 4a). Moreover, by changing the number of CDs, the CDs/polyvinyl alcohol (PVA) composite films can exhibit cyan, blue, and light-yellow emitting fluorescence. The Commission Internationale de L’Eclairage color coordinates of the composite films with the CDs concentrations of 3, 5, and 7 wt.% vary from (0.19, 0.23), through (0.25, 0.35) to (0.29, 0.40) under the UV-light excitation, respectively (Figure 4b).

#### 3.2.2. Solvation Effects (Polarity-Sensitive Fluorescence Effect)

Wang et al. [106] reported the preparation of red-emitting CDs (BNCDs), using citric acid and 1,1′-binaphthyl-2,2′-diamine via a solvothermal process. The shift of UV absorption and PL emission to a longer wavelength was observed when the polarity of solvent increased. BNCDs can be used for the sensitive detection of trace water in organic solvents because of their solvent-dependent (polarity-sensitive) fluorescence. Moreover, BNCDs can also act as a fluorescent probe for the Fe^3+^ and F^−^ sensing by using an “on-off-on” fluorescence mechanism (Figure 5).

#### 3.2.3. Size and Surface States

Hu et al. [107] found that PL properties of CDs were determined by their sizes and surface states. However, the photocatalytic activities only depend on their surface states. High upward band bending caused by surface COOH and C=O groups of CQDs leads to efficient separation of photoexcited carriers (Figure 6). The size of CDs can be tuned by changing the concentration of HNO_3_ for the oxidative refluxing process, while the surface states of CDs can be changed by selective reduction with NaBH_4_.

Zhu et al. [108] investigated the origin of the surface-related emissions and the mechanism through the surface oxidation and reduction treatment of the pristine CDs. The blue, green, and red fluorescence originated from the carbogenic core’s intrinsic state emission, the n → π* transitions of surface C=N and C=O groups, respectively. For CDs modified by the oxidation (o-CDs), the C=O groups increase, while the C=N groups disappear, owing to the oxidation by HNO_3_. On the contrary, the C=O groups in the CDs modified by the reduction (r-CDs) almost disappear due to the reduction by NaBH_4_.

#### 3.2.4. Electronic Structures and Photophysics Analysis of CDs

The synchrotron near edge X-ray absorption fine structure (NEXAFS) test is a useful tool to find out the electronic states and photophysics of materials [109,110,111]. The synchrotron NEXAFS test was utilized for the analysis of CDs-based materials. Pschunder et al. [112] studied the correlation between tunable emission performance and intricate structural features of N,B-doped CDs by the synchrotron NEXAFS tests, X-ray photoelectron spectroscopy (XPS), and steady-state and time-resolved optical spectroscopy. The simultaneous coexistence of B with N influences the electron delocalization and energy-level alignments, which eventually alter the overall photophysics of the co-doped CDs (Figure 7). The XPS analysis of N-doped CDs suggests an aromatic core domain and the N atoms containing edge. However, the core N atoms covalently bonded to neighboring atoms are predominant in the N,B-doped CDs. The N K-edge NEXAFS tests also confirm this result, revealing the significantly enhanced planar σ* resonance of the N,B-doped CDs. The co-doping of B and N inhibited the formation of green emitting fluorophores. Moon et al. [113] found that the NEXAFS results revealed that N-GCDs had a high density of graphitic structures such as sp2-hybridized carbon and tiny amounts of defect. Bokare et al. [114] studied the mechanism of forming C-dots and GQDs with different functionality and size by XPS and NEXAFS measurements. In situ X-ray absorption spectroscopy was utilized by Ren et al. [115] to analyze the electronic structures of water molecules and CDs in an aqueous dispersion. They found that the and N doping and core graphitization improved the hydrogen bonding and electron transfer ability of CDs with surrounding water molecules.

## 4. Applications of CDs

### 4.1. Degradation of Organic Toxicants

#### 4.1.1. Organic Dyes

The polluting of freshwater sources is a significant threat to providing safe drinking water. Due to unsustainably high development initiatives in construction activities, industrial emissions, and agricultural activities [116], inorganic toxicants such as heavy metals, deposits, and nutrients, and organic toxicants such as dyes, pesticides, pharmaceutical drugs, and endocrine disruptors, are deposited into the environment. The vast majority of these toxicants, mainly inorganic toxicants, will remain in the environment, leading to toxicant bioaccumulation and entry into the food matter. Finally, it has a toxic effect on higher trophic level organisms, which can be lethal. Photocatalysts have been developed for environmental cleanup in recent times [117,118]. CDs have nontoxicity, chemical inertness, photoinduced electron transfer, excellent biocompatibility, and customizable photoluminescence behavior patterns. Since CDs are frequently made from eco-friendly materials, they are inexpensive and environmentally friendly at reducing waste generation. 

CDs have better catalytic performance with enhanced photocatalytic effectiveness and shorter degradation times. CDs were made by a simple one-pot solvothermal process, using ethanol and glyoxal as a precursor (0.26). In the sunlight, indoors, and under dark conditions, the degradation efficiency of indigo carmine (IC) can reach 91% [119]. The *Cornus walteri* leaves were used to synthesize G-CDs that acted as a photocatalyst for the degradation of organic dyes [120]. A hydrothermal procedure was used to make carbon dots containing Fe and N (Fe,N-CDs) as Fenton-like catalysts for the degradation of the MB dye. About 100% of the MB dye (20 mg L^−1^) was degraded within 60 min [120]. The TiO_2_ nanoparticles decorated with a microalgae-based carbon dots (MCDs) (TiO_2_-MCDs) composite degraded the MB dye more efficiently than pristine TiO_2_. The MCDs in TiO_2_-MCDs act as the reservoirs for trapping photogenerated electrons and as photosensitizers for enhancing visible-light absorption [121]. Wang et al. [122] prepared the CDs-based porous europium micro-networks (CDs@P-Eu-MNs). The CDs can alter the morphology of CDs@P-Eu-MNs and result in a huge variety of porous structures. The incorporation of CDs can improve the activity of photocatalysts by increasing both the light absorption and the separation of photogenerated carriers. The visible-light-driven activity of the CDs/TNs (TNs-TiO_2_ sheets) photocatalysts was significantly higher than that of bare TNs [123]. The remarkable stability of CDs/TNs was explored by completing repeated Congo red (CR) degradation for five cycles (photocatalytic degradation rate reached 85.9% after irradiation for 120 min). The CDs/MoS_2_/p-C_3_N_5_ composites had outstanding photocatalytic degradation ability (93.51%) toward methylene blue. 

CQDs-modified Sb_2_WO_6_ nanosheets (CQDs/Sb_2_WO_6_) exhibited a visible-to-near-infrared (Vis-NIR) light-responsive property [124]. The photocatalytic degradation efficiency of RhB by the composite photocatalyst is roughly seven times better than that of Sb_2_WO_6_. The results of quenching experiments, DFT calculations, and electron spin resonance spectrometry revealed that the hydroxyl radical (•OH) played a dominant role in the photocatalysis reaction. Figure 8 shows the proposed photocatalytic degradation mechanism by the CQDs/Sb_2_WO_6_ photocatalyst. The conversion of the long-wavelength light from 550 to 850 nm to the short-wavelength light from 440 to 500 nm was achieved by introducing CQDs, realizing the indirect use of NIR irradiation. CQDs-loaded Sb_2_WO_6_ exhibited higher •OH production and photoexcited carrier utilization.

#### 4.1.2. Possible Mechanism

The nitrogen-doped carbon dot-modified ZnO composite (N-CDs@ZnO) photocatalyst shows significantly better activity (MB dye degradation efficiency > 99%, 60 min irradiation) than pristine ZnO photocatalysts (75%, 60 min irradiation). The improved photocatalytic activity resulted from increased UV-light absorption and hindered the recombination of photoexcited electron-hole pairs. The hindered carrier recombination may be due to effective electrons trapping by N-CDs. Moreover, loading N-CDs also helped to solve the photocorrosion problem of ZnO in the N-CDs@ZnO photocatalysts. N-CDs were decorated on the ZnO surface and created a complex structure providing the access to photogenerated charge transfer upon light irradiation. The combination of adsorbed O_2_ with the electrons in N-CDs leads to the formation of oxygen radicals. Then N-CDs with the up-conversion properties convert the long-wavelength light to the short-wavelength light that can excite ZnO to form separate electrons and holes. During the degradation process of the MB dye, the π-π* interaction between N-CDs and the MB dye can enhance the degradation of MB by the N-CDs@ZnO photocatalyst [125]. The nitrogen-rich carbon nitride (p-C3N5) had exceptional electronic properties. The CDs-modified co-catalyst MoS_2_ can significantly improve the photocatalytic activity of p-C3N5 by increasing the transfer rate of photoexcited electrons [126]. The photogenerated electrons in the CB of p-C3N5 are collected and stored by CD particles, transferred across the interface to a MoS_2_ cocatalyst, leading to enhanced separation of electron-hole pairs. The electrons transfer to MoS_2_ nanosheets and interact with the adsorbed H^+^ and evolve H_2_. The holes in the VB of p-C_3_N_5_ are consumed by the reacting with SO_3_^2−^ and S^2−^. CDs attached on MoS_2_ act as the electron acceptors, facilitating charge-transfer efficiency and improving the photocatalytic H_2_ production activity of the composite photocatalyst. Organic pollutant dyes such as RhB, MB, CR, and fuchsine were removed by using the newly developed Z-scheme C_3_N_4_-NS/CD/FeOCl ((C_3_N_4_-NS) g-C_3_N_4_ nanosheets) photocatalysts [127]. The complete removal of RhB can be achieved within 60 min. The photocatalytic activity of C_3_N_4_-NS/CD/FeOCl for the removal of RhB, CR, MB, and fuchsine was about 39.7, 15.2, 26.9, and 20.9 times that of pristine C_3_N_4_. The C_3_N_4_-NS/CD/FeOCl sample also displayed high stability. The chemical scavenging studies revealed that the •O_2_^•−^, •OH, and h^+^ played significant roles in the photocatalytic degradation reaction. The modified CDs-BiSbO_4_ composite was used for the degradation of RhB effectively up to 90% [128]. The CDs with excellent up-conversion properties can successfully convert long wavelengths (550–900 nm) to short wavelengths (320–500 nm). CDs acted as the electron sink, reducing the recombination of photogenerated carriers originating from the BiSbO_4_ nanomaterials. When O_2_ molecules are deposited on the CDs surface, the electrons in the interlayer will be drawn toward O_2_, providing a perfect platform for activating the molecular oxygen. 

#### 4.1.3. Pharmaceutical Pollutant Removal

The increasing persistence of active pharmaceutical residues in water matrices, such as anti-inflammatory medicines (diclofenac, naproxen, indomethacin, etc.) and antibiotics (tetracycline, ciprofloxacin, norfloxacin, etc.), has become a global problem [129]. These harmful substances have negative impacts on the entire living ecosystem and deplete biodiversity. As a result, the development of effective solutions and preventative strategies is urgently needed to reduce the pharmaceutically active developing contaminants from the environmental matrices in a long-term manner. As a result, CD-based photocatalysts are gaining traction as a viable decontamination option for pharmaceutical pollutants. CDs play a vital role in the performance of photocatalysts, due to their unique up-conversion property, electron transfer capabilities, and effective separation of photogenerated carriers [130].

Table 1 listed the precursors, synthesis methods, target pollutants, active species, degradation efficiencies, and roles of CDs for the removal of various organic pollutants.

### 4.2. Treatment of Inorganic Toxicant

The fabrication of CDs from microcrystalline cellulose (MCC) is described in a scalable synthetic approach [138]. Elimination of hazardous Cr^6+^ from wastewater was used to test the effectiveness of the produced CDs. Under sunshine illumination, CDs made from cellulose eliminated 20 ppm of Cr^6+^ in about 120 min. During the control test in dark surroundings, no Cr^6+^ removal was observed by the cellulose material as reference samples. With a half-life of 26 min, Cr^6+^ elimination follows pseudo-first-order dynamics. Furthermore, cyclic voltammetry research confirmed the removal of Cr^6+^ from wastewater. A photocatalytic Z-scheme TiO_2_-CDs/polyaniline electrode was constructed to improve photocatalytic activity [139]. The light adsorption was enhanced after loading carbon dots and PANI (polyaniline). Simultaneous carbamazepine degradation (44.67%) and Cr^6+^ reduction (11.94%) was observed because of the decreased bandgap and increased photocurrent density. There is a noticeable improvement in the performance of the carbamazepine degradation (from 77.63% to 83.29%) and Cr^6+^ reduction (from 23.70% to 25.68%). The free-radical •OH (78.63%) is the main active species in the degradation of carbamazepine (CBZ) into small molecules by a process consisting of hydrolysis, dehydration, de-ketonization, deaminization, and ring-opened reactions.

### 4.3. CO_2_ Reduction

CO_2_ can be utilized as a substrate for the synthesis of fuels or high-value carbon compounds such as HCOOH/HCOO^−^, CO, CH_4_, C_2_H_4_, CH_3_OH, or C_2_H_5_OH. The specificity toward the formation of certain target products is a significant obstacle for ensuring an effective reduction reaction of CO_2_, because of the possible formation of a lot of carbon-containing products, and the competitive side H_2_ evolution rate (HER), which also decreases the performance of the entire reduction reaction process of CO_2_ in aqueous media. Furthermore, the exceptional inertness and stability of CO_2_ molecules pose thermodynamic and kinetic barriers to effective CO_2_ activation and conversion. Q. Liang et al. [140] reported that, without using a sacrificial agent or any other photosensitizer, a CD-modified Co_3_O_4_/In_2_O_3_ composite catalyst for effective CO_2_ photoreduction achieves an excellent CO production activity of 2.05 mmol h^−1^ g^−1^. The CD catalyst produces 3.2 times more CO than the Ru catalyst based on the same testing conditions, namely without the addition of triethanolamine (TEOA). The transient photovoltage (TPV) measurements explored the interface charge transfer kinetics, suggesting that the CDs participate in the electron and hole transfer procedures, stabilizing charge, and enriching H^+^ for the effective reduction of CO_2_.

The oxygen vacancy defect results from a loss of oxygen atom from its relative position in the crystal lattice. The introduction of surface oxygen vacancy is a promising method to tune band structure, modify surface chemical states, and accelerate charge separation of photocatalysts. Xiong et al. [141] designed a photocatalyst GQDs/BWO_6-x_, consisting of graphene quantum dots (GQDs) and surface Vo (oxygen vacancies) with decorated Bi_2_WO_6_ (BWO). The GQDs/BWO_6-x_ showed improved photocatalytic conversion of CO from CO_2_, with a high yield (43.9 µmol g^−1^ h^−1^), i.e., 1.7 times greater than pristine BWO. The GQDs/BWO_6-x_ generated electrons showed a longer fluorescence lifetime than pristine BWO, indicating an excellent separation efficiency for photoexcited carriers. According to the DFT calculations, the electrons move to Vo-adjacent atoms from Vo-remote atoms of GQDs/BWO_6-x_. The energy barrier calculation of GQDs/BWO_6-x_ and BWO revealed that a simple transition of *COOH to *CO is the rate-limiting step. The results of DFT calculations of molecular binding energy (BE) on the surface of photocatalysts were performed to demonstrate the CO_2_ activation mechanism by spotting the adsorption energy for intermediates. The BE outlines and intermediate configurations are shown in Figure 9a,b. The CO_2_ molecules adsorbed on BWO and GQDs/BWO_6-x_ were first hydrogenated to *COOH, then converted to *CO and *OH, and then rehabilitated to *CO and *H_2_O. The *CO intermediate bonded at the Vo-neighboring sites was desorbed from the surface of GQDs/BWO_6-x_ to produce CO (possibility (1)) or hydrogenated to produce *CHO (possibility (2)). The CO desorption energy for GQDs/BWO_6-x_ is comparable to Bi_2_WO_6_. Therefore, the surfaces of GQDs/BWO_6-x_ and BWO were capable of desorbing *CO to form free CO in a similar manner. Since GQDs/BWO_6-x_ has a lower *CHO generation energy than the bulk BWO, the *CO intermediate for GQDs/BWO_6-x_ is more easily protonated to CH_4_. The formation of *CO by the protonation of *COOH is the rate-determining step. In comparison with the conversion of *COOH-BWO to *CO-BWO, the transformation of *COOH-GQDs/BWO_6-x_ to *CO-GQDs/BWO_6-x_ has a lower energy barrier, indicating that the decoration by GQDs and Vo was advantageous for the transformation of *COOH to *CO.

On a carbon nitride-like polymer (FAT) adorned with CDs, Wang et al. [142] demonstrated that CO_2_ is reduced to methanol (CH_3_OH) with 100% selectivity using H_2_O as the only electron source and discovered that carbon dots could extract holes in FAT with almost 75% efficiency before they became unreactive due to entrapment using transient absorption spectroscopy. The removal of holes resulted in a higher density of photoelectrons, indicating that shorter-lived reactive electrons recombined less frequently. Lee et al. [143] rationally created a CD/TOH hybrid catalyst consisting of N-doped carbon dots (CD) and hollow TiO_2_ spheres (TOH). They used it for the photocatalytic reduction of CO_2_ to make CH_4_. According to electron microscopy images, the CD/TOH composite has a porous hollow spherical shape that is uniformly loaded with CD. Furthermore, the CD/TOH hybrid exhibits excellent light-harvesting, large surface area, good CO_2_ adsorption capabilities, and, most critically, improved separation of photogenerated carriers for CO_2_ photoreduction processes. As a result, the CD/TOH with 2 wt.% CD exhibits a high CH_4_ generation rate of 26.8 µmol h^−1^ g^−1^, equal to 98% of CH_4_ selectivity over the competitive H_2_ generation reaction. Wang et al. [144] designed and fabricated a direct Z-scheme heterojunction composite CPDs/Bi_4_O_5_Br_2_, consisting of Bi_4_O_5_Br_2_ nanosheets and carbonized polymer dots. It effectively facilitates photogenerated carrier migration and separation efficiency while retaining more negative electron reduction potential of carbonized polymer dots (CPDs) and more positive hole oxidation potential of Bi_4_O_5_Br_2_. CPDs also facilitate the adsorption of CO_2_ and COOH* intermediates and the desorption of product CO. Under the Xe lamp irradiation, the maximum CO generation of CPDs/Bi_4_O_5_Br_2_ is 132.42 µmol h^−1^ g^−1^, which is 5.43 times higher than that of Bi_4_O_5_Br_2_ nanosheets. When the excitation wavelength is higher than 580 nm, CPDs with up-conversion capabilities can extend the light utilization range, resulting in better CO_2_ conversion performance in composite materials.

### 4.4. Hydrogen Evolution

Photocatalytic H_2_ production using semiconductor photocatalysts is an eco-friendly technology for solar energy conversion [145,146,147]. Ding et al. [148] reported the synthesis of a sequence of CDs decorated HCNS-C_x_ (hollow g-C_3_N_4_ spheres) with glucose and cyanamide as precursors. The single-step in situ thermal polymerization method was used to synthesize the HCNS-C_x_. CDs and g-C_3_N_4_ were able to maintain a tight bond and enhance the separation of photogenerated carriers. The HER of HCNS-C_1.0_ (2322 µmol g^−1^ h^−1^) was 1.8 times and 19 times higher than the HCNS and bulk g-C_3_N_4_, correspondingly. Figure 10. presents the proposed photocatalytic H_2_ production mechanism for the HCNS-C_1.0_ composite photocatalyst. CDs act as the photosensitizer that extend the light absorption wavelength of composite photocatalysts, resulting in more photoexcited carriers. CDs also facilitate the electron transport and improve the reduction of H^+^ to form H_2_.

A non-metallic photocatalyst based on CQDs/covalent triazine-based framework (CQDs/CTF) was made using an impregnation approach for photocatalytic hydrogen generation. In comparison to pristine CTF, the composite with 0.24% CQDs showed a three-fold increase in hydrogen generation rate of 102 μmol g^−1^ h^−1^. CQDs acted as the electron libraries, facilitating electron capture and promoting the separation of photogenerated carriers in CTF-1, according to photoluminescence and photoelectrochemical research. CQDs’ excitation-independent up-conversion fluorescent properties gave the catalysts a wider range of visible-light responses and improved the efficiency of solar energy utilization [149]. Wang et al. [150] reported a CQDs-CdIn_2_S_4_ (CQDs/CIS) heterostructured composite. As the number of CQDs increased, the morphology of the hybrid sample changed from 3D octahedrons to 2D nanosheets. This unique 3D/2D structure and synergistic effects between CdIn_2_S_4_ and CQDs effectively boosted the active reaction sites of the composite, improving quantum yield and photogenerated electron pair separation efficiency. In particular, the CQDs/CIS composite photocatalyst had the highest H_2_ production activity of 956.79 μmol g^−1^ h^−1^, which was 7.57 times higher than the pristine CdIn_2_S_4_. 

CQDs modified TiO_2_ composites were prepared to coproduce H_2_ and arabinose with increased selectivity. As depicted in Figure 11 when CQDs/TiO_2_ composites are exposed to light, electrons in the VB (valence band) are excited into the CB (conduction band), leaving h^+^ (holes) in the VB. The CQDs partially trap the photogenerated electrons and then cause proton reduction to produce H_2_, whereas the remaining electrons react with the absorbed oxygen to produce radical •O_2_^−^. The photogenerated h^+^ combines with the absorbed H_2_O to form radical •OH. The CQDs/TiO_2_ composites with particular colored CQDs can significantly enhance the selectivity of glucose to arabinose conversion (75%) and the H_2_ production activity (2.43 mmol h^−1^ g^−1^) [151].

Xu et al. [152] proposed that the incorporation of nitrogen-doped carbon quantum dots (NCQDs) into composite catalysts not only improved photoabsorption but also allowed effective and rapid migration of photogenerated electrons to NCQDs. The photocatalytic H_2_ generation activity can be effectively improved because of reduced photogenerated carrier recombination. The photocatalytic H_2_ generation performance of composite photocatalysts was significantly enhanced (2306.1 µmol g^−1^ h^−1^), which was around seven times that of the sample without NCQDs. 

### 4.5. Antimicrobial

#### 4.5.1. Food Storage

The curcumin (Cur) carbon quantum dots (Cur-NRCQDs), as a photosensitizer, can improve the efficiency of reactive oxygen generation and antibacterial performance. Cur-NRCQDs can inactivate 100% Escherichia coli (*E. coli*) and Staphylococcus aureus (*S. aureus*) under xenon lamp irradiation at concentrations of 10 and 15 M, respectively. The reactive oxygen created by Cur-NRCQDs during photodynamic therapy may have disrupted the cell membrane, leading to leakage of the contents [153]. Lin et al. [154] prepared nanosized, spherical, neutral charge, fluorescent carbon dots with good water dispersibility using fish, ginger, onion, and garlic as carbon sources. The fish and ginger CDs contained lower sulfur elements than the onion and garlic CDs. The onion CDs exhibited antibacterial activity against *Pseudomonas fragi* (*P. fragi*), as well as antimicrobial activity against *S. aureus* and *E. coli*. Onion CDs had MIC of 2 mg mL^−1^ and MBC of 4 mg mL^−1^, against *P. fragi*. The cell membrane and cell-wall integrity were damaged after the light irradiation to CDs, and extracellular alkaline phosphatase (AKP) and ATP activity increased, resulting in a decrease in cell viability and a change in cellular shape in *P. fragi*. These results reveal that onion CDs can be used as a bacteriostatic agent for aquatic products. Ma et al. [155] produced AgNPs by reducing a combination of CDs and silver (Ag) ions with sodium borohydride (NaBH_4_), employing CDs with numerous chemical groups as ligands. The stable CD-AgNPs (carbon-dot-stabilized silver nanoparticles) give a higher antibacterial performance than AgNPs without CDs. The CD-AgNPs had an MIC of 100 µg mL^−1^ with 0.613 µg mL^−1^ Ag. CD-AgNPs have a broad-spectrum antibacterial activity since they inhibit the growth of six bacteria, namely *S. aureus*, *L. monocytogenes*, *E. coli*, *Vibrio parahaemolyticus*, *S. typhimurium*, and *Shigella castellani*.

#### 4.5.2. Wound Healing

The quaternized carbon quantum dots (qCQDs) regained the weight of rats in wounds infections caused by mixed bacteria, significantly decreased the death of rats from a serious infection, and also improved the healing of infected wounds in rat models. Biosafety tests revealed that qCQDs had no apparent toxic or adverse reactions during the testing phase. The quantification proteomics analysis showed that qCQDs primarily acted on ribosomal proteins in S. aureus and downregulated the citrate cycle proteins in *E. coli* [156].

Wu et al. [157] reported that the levofloxacin (antibiotic medicine)-based carbon dots (LCDs) improved the antibacterial activity and reduced drug resistance. The results showed that LCDs have effective antibacterial properties due to the active groups of levofloxacin being preserved. LCDs had a dual-antibacterial mode that distorted microorganisms because of the reactive species and positive surface charge generation at the same time. The LCDs in vivo antibacterial effects were evaluated in the infected wounds with *E. coli* or *S. aureus* of ICR mice (Figure 12). The abscess was visible in infected wounds after 48 h of infection with *E. coli* or *S. aureus*. When the infected wounds were treated with levofloxacin hydrochloride, LCDs, and normal saline, from day 1 to day 10, as illustrated in Figure 12. The scab appeared after treatment in the LCDs and levofloxacin hydrochloride (LC-HCl) groups, but in the negative control group, the exudates and pus remained. The exudates and pus remained in the negative control group after seven days of treatment, although the area of infected wounds in the LCD groups was substantially reduced compared to the LC-HCl groups.

The sulfur (S) functionalized turmeric-derived carbon dots (S-CDs) showed high antibacterial and antioxidant action against mouse fibroblast L929 cells with minimal cytotoxicity [158]. The UV protection capabilities of the CDs-added film were increased without affecting the transparency of the pectin/gelatin film. The hydrophobicity, mechanical, and water vapor permeability of the film were all changed by the addition of CDs. Furthermore, the DPPH and ABTS techniques revealed that CDs-loaded pectin/gelatin films had high antioxidant properties. Moreover, the film with sulfur functionalized CDs demonstrated high antibacterial action against foodborne pathogenic bacteria such as *E. coli* and *L. monocytogenes*. S-CDs embedded pectin/gelatin films can be employed in food packaging-related applications to increase the shelf life of foods and assure food safety. Glucose carbon dots (GCDs) were made by utilizing glucose as the carbon source and doped with sulfur, nitrogen, and boron to improve their functioning [159]. The NGCD, in particular, has the highest antioxidant activity of all the GCD. The S-doped GCD (SGCD) and B-doped GCD (BGCD) showed higher antibacterial activity against *L. monocytogenes* and *E. coli*. Both bacterial strains are susceptible to the NGCD, which has the strongest antibacterial action. The NGCD possesses substantial antifungal activity against *P. citrinum*, *Ammophilus fumigatus*, *R. rubra*, and *C. albicans*, but the SGCD effectively inhibits the *F. solani* growth. When mouse fibroblast L929 cells are exposed to a high dosage of 500 g/mL for 72 h, 80% of cells remained alive, indicating the low toxicity of CDs. The Ag,NCQDs are efficient antimicrobials against *E. coli* and *S. aureus* [160]. Antibacterial activity was evaluated by using *S. aureus* and *E. coli* colonies, and the morphologies of the bacteria were investigated by using SEM. The inhibitory effect is sequenced in spread plate tests: NCQDs > Ag,NCQDs > Ag-CQDs > NCQDs > Ag solution. As a result, doping Ag into N-CQDs is a more effective technique to improve antibacterial activity than mixing N-CQDs and Ag. Furthermore, Ag,NCQDs have antibacterial activity against both bacteria colonies, whereas the NCQDs counterpart has an inhibitory effect solely on *E. coli*. The lowest Ag,NCQDs preventive concentrations against *E. coli* and *S. aureus* are 250 and 200 µg mL^−1^, respectively.

### 4.6. Cell Imaging

The development of fluorescent tools can improve the ability to probe biological dynamics [161]. The CDs-HS18 was employed as the fluorescent dyes for cell imaging study in the *Saccharomyces cerevisiae* (*S. cerevisiae*) bacteria, *Jasminum mesnyi* Hance (JMH) leaf, onion skin, and microworms via in vivo and in vitro methods, respectively [162]. The fluorescence signals found in JMH are in the leaf veins, revealing the supply tendency to the tip from the petiole, which is constant with the movement of H_2_O in the leaves, indicating the distribution of CDs in the leaves is mainly driven by transpiration. The confocal laser scanning microscopy (CLSM) data show that multicolored fluorescent signals can be found in the pharynx, somatic cells, pseudocoelom, and digestive system of the microworms. However, the signal strength in the mouth cavity and the tail is lower than in other organs. Wang et al. [162] assume that CDs are ingested by microworms by ingestion. Due to the exceptional biocompatibility of CDs, CDs enter the gastrointestinal tract through the oral cavity and pharynx, spreading to the somatic cells and pseudocoelom, and eventually being expelled through the anus. Microworm deaths were not seen during the test, implying that the regular metabolic activities of microworms were maintained.

Figure 13 presents the synthesis of Ca-, N-, and S-doped CDs (Mis-mPD-CDs) from plant Miswak and mPD and their application for cell-imaging and intracellular CR sensing. The negatively charged and green-emitting Mis-mPD-CD can enter and label bacteria, fungi, and animal cells for long periods with excellent biocompatibility and stability. The Mis-mPD-CDs were used to detect and image CR in living cells (*C. albicans*, A549, *S. aureus*, and *E. coli*) and zebrafish. Mis-mPD-CDs were proved to be useful for the quantitative sensing of CR in real samples of industrial wastewater and fish tissues, as well [163].

The level of folic acid in the human body can be a useful indicator for evaluating the body’s normal physiological activities and can provide information about cell growth and reproduction [164]. A high level of FA can cause a variety of diseases. The fluorescence spectra of N-CDs were quenched after the addition of folic acid due to the synergistic effects of the static quenching mechanism and internal filtering effect (IFE). Within the folic acid concentration range of 0–200.0 M, the LOD was 28.0 nM (S/N = 3) under optimal conditions. Furthermore, N-CDs were used to sense folic acid in the real samples, including urine and fetal bovine serum, with a quantitative addition recovery rate of 99.6–100.7%. The experimental results revealed that N-CDs exhibit low toxicity, excellent cell imaging performance, and quantitative folic acid analysis. In the daylight, aqueous dispersions of Morus nigra CDs (M-CDs) exhibit a brownish-yellow color and cyan-blue light emission under UV light irradiation [165]. M-CDs show characteristic excitation-dependent emission with a high QY of 24%. M-CDs exhibit a high QY of 24% in a characteristic excitation-dependent emission manner. M-CDs were used as fluorescent sensors for the sensitive and selective Fe^3+^ sensing via fluorescence quenching, with a detection ranging from 5 to 30 µm, and an LOD of 0.47 µM. Furthermore, M-CDs were applied to stain human colon cancer (HTC-116) cells for cell viability and microscopic analysis. The M-CDs-conjugated HTC-116 cells emitted blue, green, and red light when excited through 405, 488, and 555 nm filters, respectively. An efficient and environmentally friendly process for the preparation of nitrogen-doped carbon dots (N-CDs) was reported, using chicken waste Galli Gigerii Endothelium Corneum (GGEC) as a precursor [166]. Surprisingly, N-CDs can be used as a sensor for the selective and sensitive detection of nitroimidazoles, such as metronidazole, tinidazole, ornidazole, and secnidazole, using internal filtration effect (IFE) and static quenching mechanisms. N-CDs were proved to be effective for detecting nitroimidazoles in some real samples (e.g., chicken, plasma, and tablets). Moreover, N-CDs also showed great potential in the bioimaging application. The Kiwi-fruit-peel carbon dots (KFP-CDs) were successfully prepared from kiwi fruit peels without using a capping/passivation agent [167]. KFP-CDs were useful for the fabrication of novel fluorescent inks. When exposed to UV light, the images and words were instantly visible. Furthermore, when evaluated for the cell-imaging application in human cell lines, KFP-CDs are biocompatible and have low cytotoxicity. The findings suggest that KFP-CDs can be used as a cell labeling agent for the in vitro imaging of cancer cells and normal cells. 

Most CDs that can discriminate between live and dead cells exhibit excitation-wavelength-dependent fluorescence and low photoluminescence quantum yields [168]. It may cause problems such as possible fluorescence overlap with the other fluorescent probe. Meanwhile, it is not feasible for dual-color live/dead staining. Therefore, developing CDs with high photoluminescence quantum yields and excitation-wavelength-independent emission becomes an important task. Excitation-wavelength-independent sulfur-doped CDs (S-CDs) were used to demonstrate that the S-CDs could distinguish dead cells from live cells for fungal, bacterial, and animal cells. Yu et al. [168] found that S-CDs could reach the interior of the dead cells, allowing the visualization of these cells. On the other hand, live cells cannot be stained by S-CDs because S-CDs cannot enter the live cells. Moreover, compared with the commercial live/dead staining dye propidium iodide, S-CDs showed better photostability and biocompatibility, indicating a promising future in cell viability assessment and cell-imaging applications. 

### 4.7. CDs for Pollutant Sensing 

Compared with conventional analysis techniques, fluorescence-based sensing has attracted attention because it has some advantages, including great sensitivity, rapid response time, easy operation, low cost, and efficiency [169,170]. Among the various fluorescent carbon nanomaterials, CDs have captivated scientists’ interest due to their ability to combine the essential selective receptors on their surface. Due to the ease with which CDs can be surface modified to employ a wide range of intrinsic functional groups, they can rapidly improve their selectivity against specific targets. Theoretically, any fluorescence changes attributed to the concentration of various analytes, such as wavelength, intensity, anisotropy, or longevity, have the possibilities to be used as sensors. 

Ji et al. [171] studied the N-doped carbon dots mediated by cetyltrimethylammonium bromide (CTAB/NCDs). The spatial structure created by CTAB/NCDs can selectively collect Hg^2+^ and cause fluorescence quenching by interacting with the surface functional groups of NCDs. The N-doped carbon dots (NCDs) were made in a hydrothermal synthesis. The NCDs were discovered to offer potential as a fluorescence sensor for detecting Hg^2+^ [172]. Static quenching of NCDs by Hg^2+^ could be a viable detection technique. The findings suggested that an NCDs-based sensor could detect Hg^2+^ in a real beverage sample. Zhang et al. [173] reported that red fluorescent InP/ZnS quantum dots (InPQDs), MOFs (ZIF-8), and blue-fluorescent carbon dots (CDs) were merged into nanosensor CDs/InPQDs@ZIF-8 for the successive optical detection of Hg^2+^, utilizing an in situ synthesis approach. The nano-low-sensor’s detection limit and its high specificity and accuracy meet the requirements for safe Hg^2+^ regulating and checking in environmental and drinking water. Furthermore, color recognition and processing software put on a smartphone may detect Hg^2+^ in real time and at a high rate. The CDs doped with Eu^3+^ ions (Eu-CDs) were produced hydrothermally, utilizing citric acid and urea as precursors and Eu(NO_3_)_3_ as a europium source by Correia et al. [174]. The Eu^3+^ ions are strongly linked to the carboxylate groups on the CDs’ surfaces and incorporated into the carbon core’s nanographene network. CDs doped with Eu^3+^ have higher diameters than CDs that are not doped, but they are split into smaller sp^2^ carbon domains. Hg^2+^ and Ag^+^ greatly quench the CDs’ luminescence, but other cations had no effect. Depending on the ion’s nature, the quenching mechanism varies greatly. The blue emission of CDs is affected by the presence of Ag^+^. In the case of Hg^2+^, the blue emission of CDs and the red emission of Eu^3+^ are quenched. Shen et al. [175] used Shewanella oneidensis MR-1 to make fluorescent carbon dots (CDs@MR-1) via a hydrothermal procedure to detect Hg^2+^ and tetracycline water samples. The limits of detection for Hg^2+^ and tetracycline were 0.43 and 0.21 µg·mL^−1^, respectively. The internal filtration effect (IFE) causes the fluorescence quenching for tetracycline. On the other hand, along with IFE, dynamic quenching and static quenching mechanisms are involved in detecting Hg^2+^. The CDs@MR-1 can also identify Gram-positive bacteria from Gram-negative bacteria (Figure 14).

An Fe/N-doped CDs (CDBFe) catalytic amplification Apt method was developed for sensitive and fast SERS/RRS/Abs tri-mode sensing of ultra-trace Pb(II) ions [176]. There is an excellent linear relationship between the SERS intensity and Pb(II) concentration between 1.3 and 16 pM (picomole). Moreover, As(III) and Hg(II) can also be monitored by this assay. A dual-emission ratio fluorescent sensor, N-CDs/R-CDs@ZIF-8, was prepared by mixing N-doped blue fluorescence carbon dots and red fluorescence carbon dots in the environment of ZIF-8 [177]. The addition of Pb^2+^ has a great effect on inhibiting the fluorescence of N-CDs, while showing little influence on the fluorescence of R-CDs. As a result, N-CDs/R-CDs@ZIF-8 can act as a ratiometric fluorescence probe for Pb^2+^ sensing. The functionalized-GQD (F-GQD) was prepared by the edge functionalization of graphene quantum dots (GQDs) by 2,6-diaminopyridine molecules and served as a fluorescent nanoprobe [178]. The F-GQD is a suitable pH sensor in the 2–6 pH range because its fluorescence is very sensitive to the pH of the medium. With the addition of Pb^2+^, F-GQD exhibits fluorescence turn-on performance. The fluorescence enhancement could be due to nanodot aggregation caused by Pb^2+^. A fluorescent NCDs (nitrogen-doped carbon dots) sensor can detect the presence of divalent heavy metal ions. The detection limits of Pb^2+^ and Cu^2+^ can reach 3 and 15 ppb, respectively [179]. Additionally, the UV absorbance spectra of NCDs were used to detect Pb^2+^ in a dose-dependent manner. 

Cr^6+^ is a toxic heavy metal that bacteria cannot decompose or consume. As a result, the development of a highly selective and sensitive method to detect Cr^6+^ is urgently needed. Wang et al. [162] designed the fluorescence CDs-HS18, prepared from Ureibacillus thermosphaericus, using a hydrothermal method. The CDs-HS18-based fast and selective sensing platform displayed an outstanding linear relation for Cr^6+^ between 0 and 9 µM, with a limit of detection of 36 nM. A majority of reported Cr^6+^ sensors are based on the fluorescence quenching method, whereas the one reported by Chen et al. [180] is performed by the fluorescence enhancement mechanism. Results of TEM, FTIR, X-ray photoelectron spectroscopy, and XRD analysis revealed that Cr^6+^ could increase the probe’s fluorescence. The detection phenomenon was established by conferring the affiliation between the quenching efficiency of CDs-Kan fluorescence intensity and the concentration of Cr^6+^ [181]. 

N-and-S-co-doped carbon dots (NSCDs) can act as fluorescent probes to detect Cr^3+^. These NSCDs exhibited strong fluorescence that was quickly quenched by Cr^3+^ even in the presence of other competing ions, demonstrating that NSCD probes have outstanding selectivity and anti-interference ability. The mechanism of fluorescence quenching of NSCD by Cr^3+^ ion may result from the increased non-radiation emission due to the deteriorated NSCD surface. These non-toxic and biocompatible NSCDs could be used to detect Cr^3+^ in living cells [182]. An orange-emission carbon dots (OCDs) were created by using a hydrothermal method. According to TEM analysis, the new OCDs were similar in size, with an average particle size of 4–7 nm [183]. The fluorescence of OCDs was effectively quenched by Cr^3+^. Notably, OCDs were effectively used for cell-level fluorescence detection of Cr^3+^ ions. N-doped carbon quantum dots (N-CQDs) were created by a top-down method, i.e., •OH radical opening of fullerene with H_2_O_2_ in a basic environment with ammonia for two different reaction times [184]. N-CQDs were tested for metal ion detection in aqueous solutions and during bioimaging. They exhibited a shift in Cu^2+^ and Cr^3+^ selectivity at a greater extent of -NH_2_ functionalization.

The copper-doped CDs (Cu-CDs) prepared by a pyrolysis approach were found to have peroxidase-mimicking activity, which reduced the fluorescent intensity of the Cu-CDs-mediated o-phenylenediamine (OPD) oxidized to a fluorescent 2,3-diaminophenazine (DAP) in the presence of H_2_O_2_ via the generation of •OH radicals. The Cr^6+^ species were reduced to Cr^3+^ by using the reductant 2,2′-azino-bis (3-ethylbenzothiazoline-6-sulfonate) (ABTS). Moreover, the fluorescent intensity of DAP is proportional to the Cr^3+^ content within the range of 5 × 10^−6^ to 1.5 × 10^−4^ M. It can be used to determine Cr^6+^ and Cr^3+^ levels in water samples (Figure 15) [185].

Fluorescent CQDs synthesized from crab-shell waste by a hydrothermal process exhibited excellent Cd^2+^ sensing and antibacterial activity. CQDs exhibited bright green fluorescence and exorbitant quenching of Cd^2+^ ions in aqueous media under UV light irradiation [186]. Wang et al. [187] reported gold nanoclusters (AuNCs) as fluorescence molecules with red emission, while N-and-S-co-doped CDs (N,S-CDs) were reported as signaling molecules. The fluorescence quenching is caused by the fluorescence resonance energy transfer (FRET) effect between N,S-CDs and AuNCs. The added Cd^2+^ can interact with AuNCs to form complexes and interact with the O atoms in the carboxyl and hydroxyl groups of N,S-CDs, resulting in a stable complex, causing N,S-CDs/AuNCs to aggregate, and then resulting in enhanced fluorescence. The addition of Cd^2+^ led to increased fluorescence intensities of N,S-CDs (F435) and AuNCs (F630). 

Wu et al. [188] synthesized green fluorescence CDs (GCDs) by using 1,4-dihydroxyanthraquinone. The GCDs based sensor is used as a dual-mode visual sensor based on “off-on” fluorescence analysis to detect Cu^2+^ and glyphosate. Turn-off fluorescence was observed when adding Cu^2+^ to GCDs, indicating a strong binding interaction between GCDs and Cu^2+^ ions. While turn-on fluorescence was observed after adding glyphosate to GCDs/Cu^2+^, these studies suggest that Cu^2+^ has more binding sites to glyphosate than other competing pesticides. A GCDs-based smart-sensing membrane was applied to detect glyphosate on vegetable surfaces. The QY of NCDs can be elevated by liquid-liquid extraction and purification. NCDs have the potential to be an excellent multifunctional sensing platform for Cu^2+^ and tetracycline antibiotics (TCs) such as oxytetracycline (OTC), tetracycline hydrochloride (TC), demeclocycline hydrochloride (DMC), doxycycline hydrochloride (DC), and minocycline hydrochloride (MC). Compared to other coexisting metal ions and antibiotics, this platform responded differently to Cu^2+^ and TCs. IFE and static quenching were identified as the fluorescent quenching mechanisms of Cu^2+^ and TCs to NCDs. The platform was successfully applied to detect Cu^2+^ and TCs in real samples, including water, milk, and urine [189]. The precursors, synthesis methods, quantum yields, target analytes, linear ranges, and detection limits of various CDs for the sensing applications are summarized in Table 2.

### 4.8. Possible Applications and Roles of Various CDs

Up to the present, the possibility of the use of CDs in a wide range of applications has been reported, including bioimaging, catalysis, energy, sensing, therapy, fertilizer, separation, security authentication, food packing, and flame retardant. Due to the unique structure, adjustable chemical, physical, optical, and electronic properties of CDs, they can play different roles in diverse applications. The roles of CDs in these applications are also summarized in Table 3.

## 5. Summary and Future Perspective

The unique properties of CDs, such as good biocompatibility, high photostability, excellent light-harvesting, up-conversion, effective electron transfer, and bandgap narrowing, make the CDs promising nanomaterials for applications in many fields. This review mainly focused on the recent progress of CDs in bioimaging, sensing, therapy, energy, fertilizer, separation, security authentication, food packing, flame retardant, and co-catalyst for environmental remediation applications. In addition, key factors affecting the properties and performance of CDs were listed in this review. The properties and performance of CDs can be tuned by some synthesis parameters, such as the incubation time and precursor ratio of the microwave-assisted process, the laser pulse width of the laser ablation synthesis, and the average molar mass of polymeric precursor. Surface passivation also has a significant influence on the particle size of CDs. Moreover, some factors affect the properties and performance of CDs, such as the polarity-sensitive fluorescence effect and concentration-dependent multicolor luminescence, together with the size and surface states of CDs.

Future efforts about CDs-based nanomaterials can focus on improving their fabrication, purification, and characterization technology, as well as expanding versatile applications and realizing structure/composition/property correlation and basic fluorescence mechanism. Some opportunities and challenges for future study are listed as follows.

(1)Commercial-scale fabrication:

A lot of researchers have been devoted to the study of CDs and expanding their use in a wide range of applications. However, the complicated synthesis process and low yield hinder the commercial-scale fabrication and industrial application of CDs. Developing robust large-scale synthetic methods for high-quality CDs is critical to their applications. Although some progress in the fabrication of kilogram-scale CDs has been reported in the past three years (see Section 2.1.3), the scale-up synthesis of CDs still faces some challenges, such as low-cost synthesis procedures, batch-to-batch consistency, and efficient purification techniques. In the future, we should develop low-cost synthesis procedures, efficient purification techniques, and novel technology for reducing the by-products and improving the reproducibility of CDs concerning their size and quantum yield. 

(2)Surface functional groups:

During the synthesis, various functional groups (amine, carboxylic acid, and hydroxyl groups) can be generated on the surface of CDs. The appropriate control of these functional groups is quite important for elevating the sensing and catalysis performance of CDs-based materials. For example, appropriate surface functional groups support the efficient attachment of CDs co-catalyst on the catalyst’s surface, thereby providing more active sites for improving the photocatalytic activity of the composite photocatalyst. Compared with bare CDs, functionalized CDs with excellent fluorescence properties and various functional groups exhibit better selectivity and sensitivity for detecting specific analytes. In the future, robust surface modification to generate reproducible surface functional groups on CDs will be a crucial factor for the real-world application of CDs.

(3)Structure/composition-property correlation:

Specific hurdles should be overcome for the application of functional CDs in different fields. For example, fluorescent sensing needs CDs-based probes with a high quantum yield. Biocompatibility is the primary requirement for nanomedicine applications. The good electrical conductivity and morphology of CDs and close contact between CDs and catalysts are essential for catalytic applications. Since the certain property of CDs is critical for some specific applications, it is important to tune the CDs’ properties. Therefore, exploring the composition-property correlation or the key factors for certain properties of CDs will be an important topic for future work. 

(4)Photoluminescence property:

Stable fluorescence is required for the LED-based application. Environment-dependent or analyte-dependent fluorescent response is critical to sensor applications. The development of CDs with long-wavelength-emitting properties and up-conversion luminescence is significant for exploring useful sensors. Hence, the mechanism of photoluminescence, the technology for tuning the light-emissive wavelength of CDs, and the improvement of the fluorescence quantum yield should be studied in the future.

(5)Characterization techniques:

It was reported that the existence of CDs in most cases could not be confirmed by XRD spectra, due to the low crystallinity and high dispersion of CDs [8]. In the future, the use of appropriate characterization methods is quite important for the research of CDs-based materials. For example, the presence of CDs in the photocatalyst composites can be determined by measuring their chemical compositions (by FTIR and XPS), morphology (TEM), and properties (UV-Vis spectra, photoluminescence spectra, photocurrent, and electrochemical impedance spectroscopy). Moreover, the synchrotron NEXAFS test is a useful tool to find out the electronic states and photophysics of CDs-based materials (Section 3.2.4). The correlation between emission performance and intricate structural features of doped CDs can be analyzed by the synchrotron NEXAFS tests, X-ray photoelectron spectroscopy, and steady-state and time-resolved optical spectroscopy [112].

## Figures and Tables

**Figure 1 polymers-14-02153-f001:**
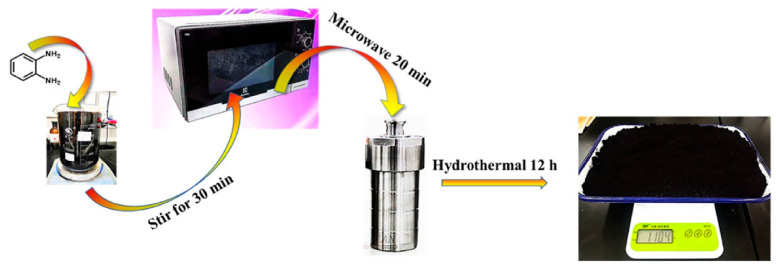
The synthesis process of C-dots_606_ from *o*-PDA at a kilogram scale (1.104 kg) with high yield by a combined microwave-hydrothermal process (reprinted with permission from [50], 2022, Elsevier).

**Figure 2 polymers-14-02153-f002:**
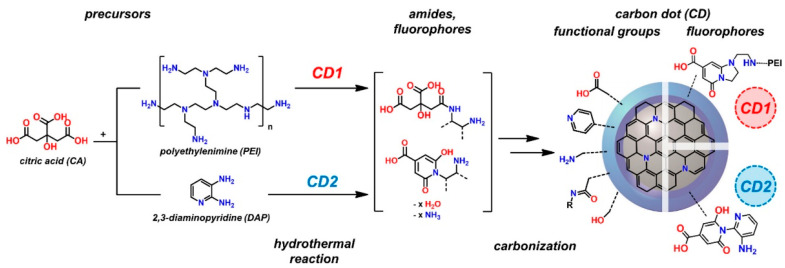
The preparation of CD1 and CD2 by the solvothermal pyrolysis of citric acid and amine precursors PEI and DAP, respectively (reprinted with permission from [94], 2020, American Chemical Society).

**Figure 3 polymers-14-02153-f003:**
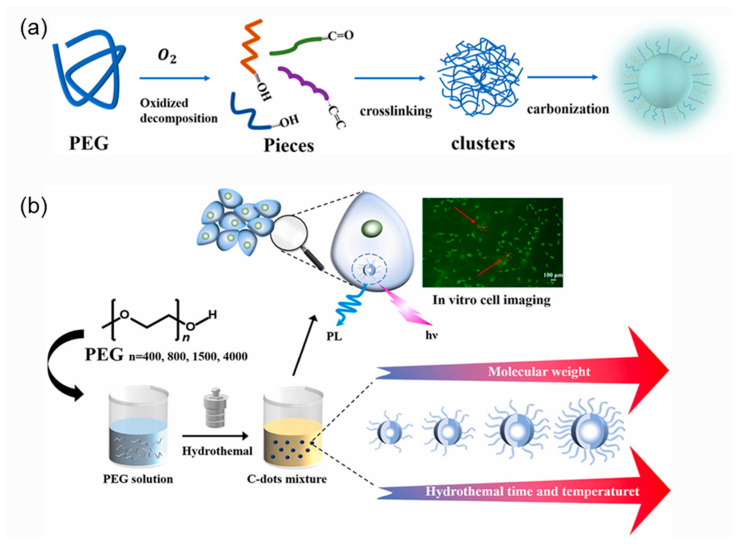
(**a**) Schemes for the preparation of CDs from PEG precursors with different MW. (**b**) The molecular weight of PEG precursors had a large influence on the fluorescence, size, surface chemistry, bioimaging performance, and thermal stability of CDs (reprinted with permission from [98], 2022, Elsevier).

**Figure 4 polymers-14-02153-f004:**
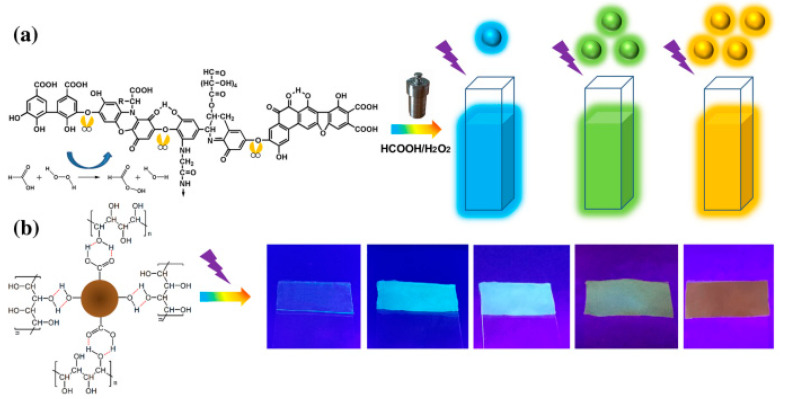
(**a**) The fabrication of CDs that exhibited concentration-dependent multicolor luminescence in deionized water, N,N-dimethylformamide, and formic acid (**b**) concentration-dependent multicolor CDs/PVA composite films exhibiting cyan, blue, and light-yellow emitting fluorescence (reprinted with permission from [105], 2022, Springer Nature).

**Figure 5 polymers-14-02153-f005:**
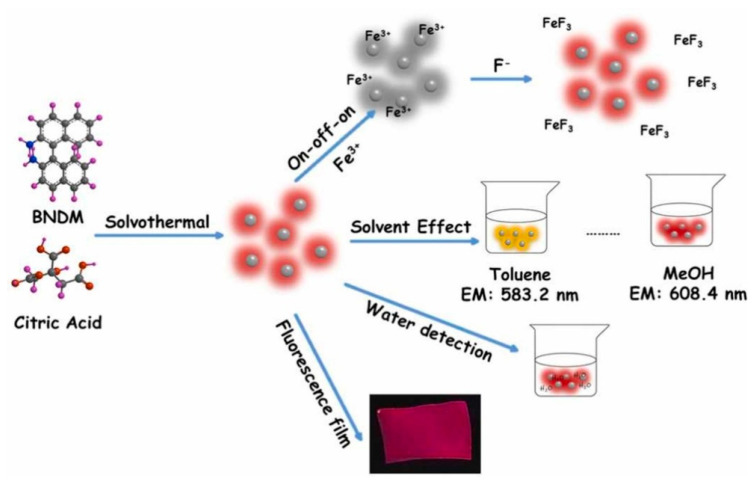
The polarity-sensitive fluorescence effect and Fe^3+^/F^−^ sensing by “on-off-on” fluorescence mechanism (reprinted with permission from [106], 2022, Elsevier).

**Figure 6 polymers-14-02153-f006:**
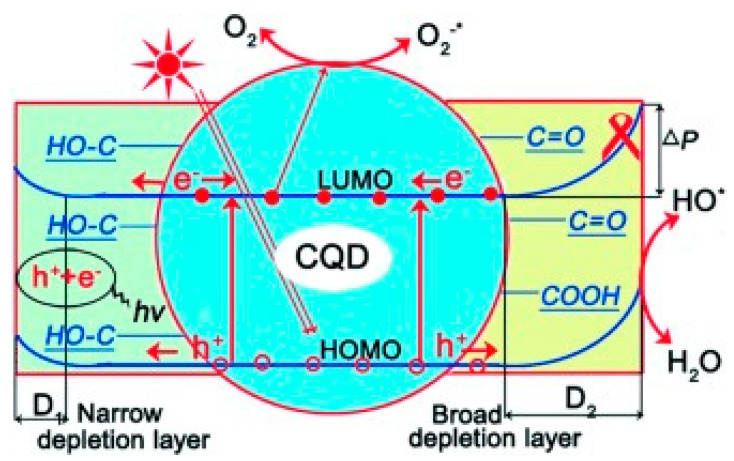
The effects of sizes and surface states of CQDs on the photoluminescent behavior and photocatalytic activities (reprinted with permission from [107], 2013, John Wiley and Sons).

**Figure 7 polymers-14-02153-f007:**
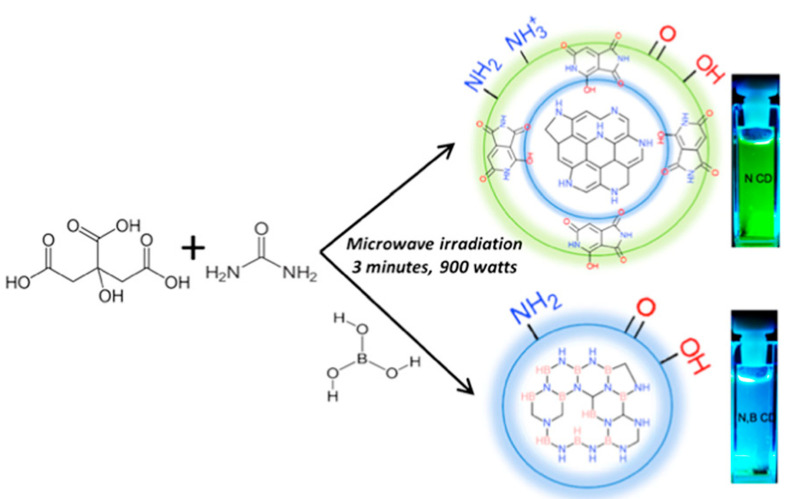
The synthesis of N-CDs and NB-CDs and the corresponding photographs under UV-light irradiation (reprinted with permission from [112], 2019, American Chemical Society).

**Figure 8 polymers-14-02153-f008:**
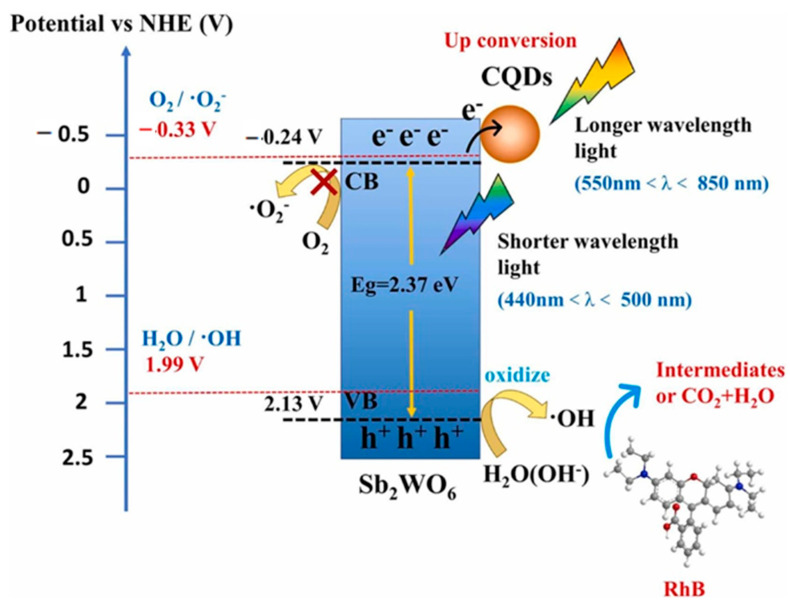
Proposed photocatalytic degradation mechanism by the CQDs/Sb_2_WO_6_ photocatalyst (reprinted with permission from [124], 2022, Elsevier).

**Figure 9 polymers-14-02153-f009:**
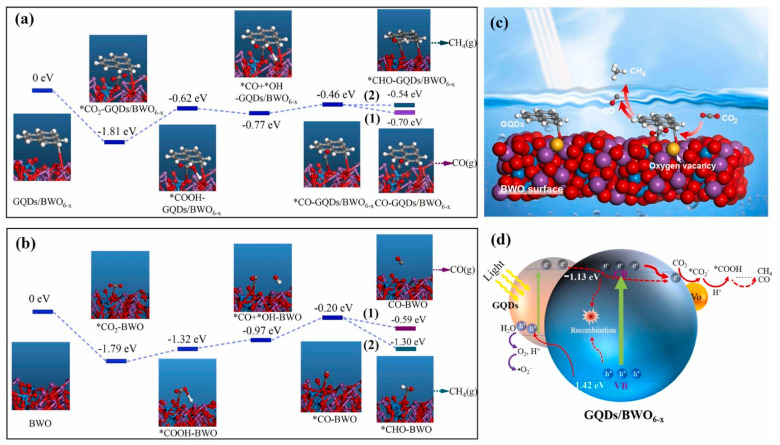
The free energy diagram for reducing CO_2_ to *CHO or CO by (**a**) GQDs/BWO_6-x_ and (**b**) BWO. (**c**) Possible microstructure of GQDs/BWO_6-x_. (**d**) Proposed photocatalytic CO_2_ reduction mechanism by GQDs/BWO_6-x_ (reprinted with permission from [141], 2022, Elsevier).

**Figure 10 polymers-14-02153-f010:**
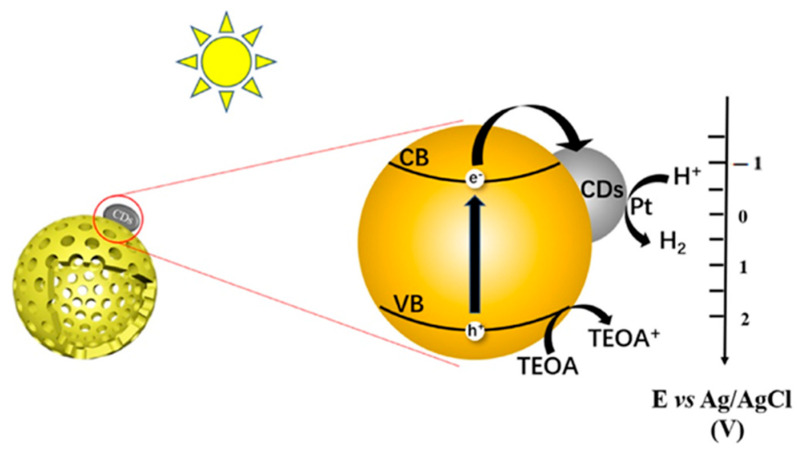
The proposed photocatalytic H_2_ production mechanism for the HCNS-C1.0 photocatalyst (reprinted with permission from [148], 2022, Elsevier).

**Figure 11 polymers-14-02153-f011:**
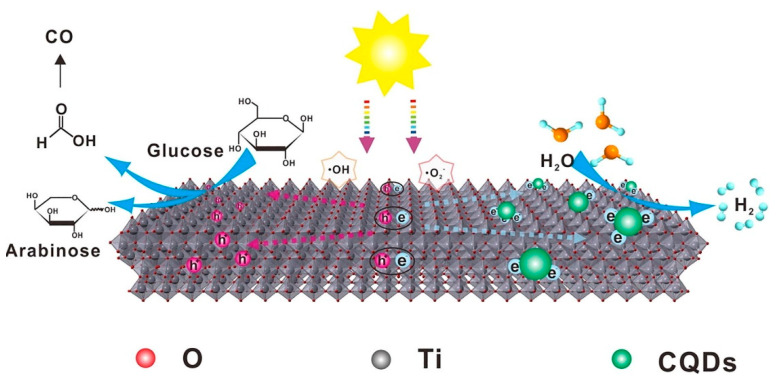
Proposed reaction mechanism for glucose photoreforming (reprinted with permission from [151], 2022, Elsevier).

**Figure 12 polymers-14-02153-f012:**
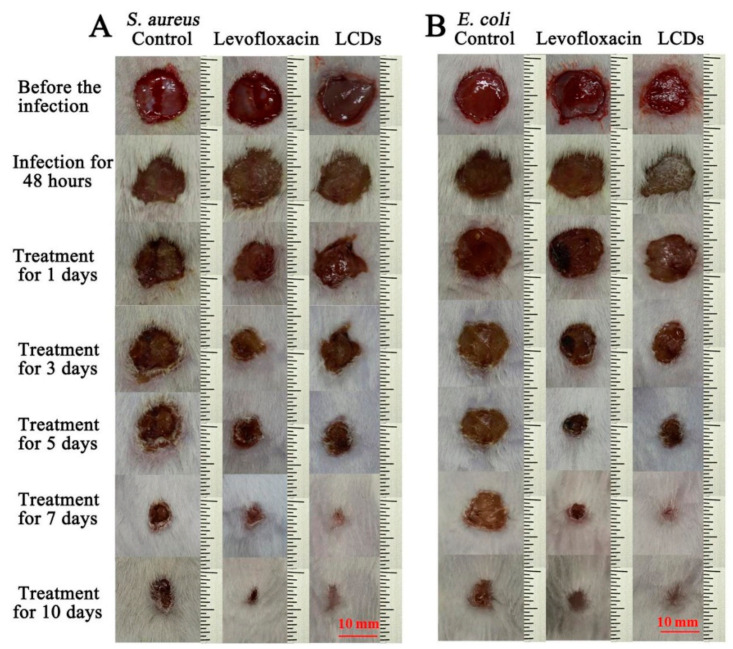
Photographic images of the wound healing progress (1 to 10 days) of (**A**) the infected wounds with *S. aureus* and (**B**) infected wounds with *E. coli* treatment with normal saline, LC-HCl, and LCDs (reprinted with permission from [157], 2022, Elsevier).

**Figure 13 polymers-14-02153-f013:**
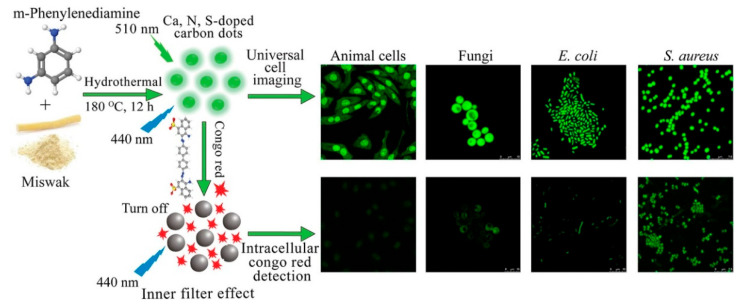
Synthesis of Mis-mPD-CD and their application for cell-imaging and intracellular CR sensing (reprinted with permission from [163], 2022, Elsevier).

**Figure 14 polymers-14-02153-f014:**
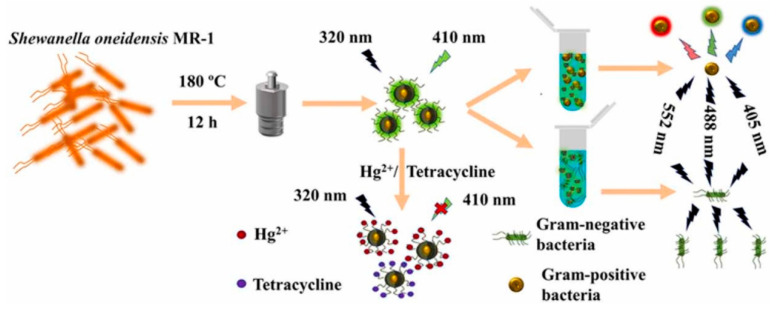
The synthesis of CDs@MR-1 and its application to identify Gram-positive bacteria from Gram-negative bacteria (reprinted with permission from [175], 2022, Elsevier).

**Figure 15 polymers-14-02153-f015:**
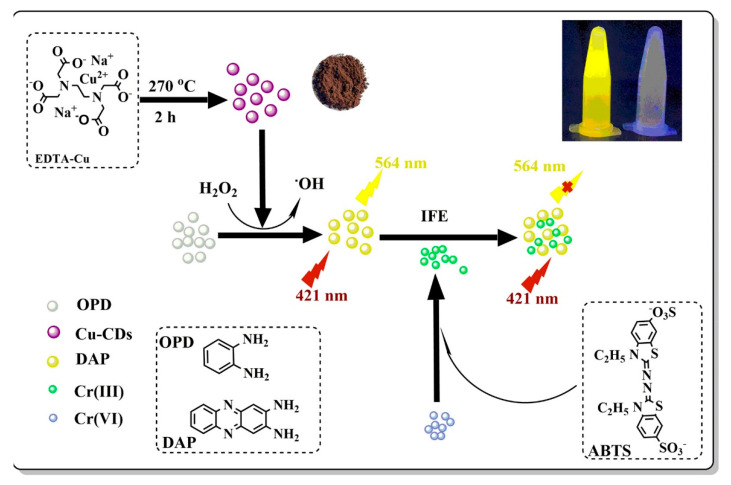
Spectrofluorimetric detection of Cr(III) and Cr(VI) based on the peroxidase-mimicking activity of Cu-CDs (reprinted with permission from [185], 2022, Elsevier).

**Table 1 polymers-14-02153-t001:** Precursors, synthesis methods, target pollutants, active species, degradation efficiencies, and roles of CDs for the removal of various organic pollutants.

Type *	CDs-Based Photocatalyst	CDs Synthesis Method	CD Precursor	Pollutant Removed	Main Active Species	Role of CDs	Efficiency (%/min of Irradiation)	Reference/Year
1	CDs	Solvothermal	Glyoxal and ethanol	Indigo carmine (IC)	•O_2_^•−^, h^+^, •OH	e^−^-h^+^ pair separation	91/4.5	[119] 2022
1	CDs	Carbonization	Bitter apple peel	Crystal violet (CV)	h^+^, •OH	efficient e^−^/h^+^ separation	100/90	[21] 2020
1	CNDs	Carbonization (pyrolysis)	Olive solid wastes	MB	O_2_^•−^ •OH	e^−^-h^+^ pair separation	100/120	[19] 2022
1	CQDs	Stirrer-assisted	Muskmelon peel	RhB	•OH	Up-conversion charge separation	99.11/35	[23] 2022
1	G-CDs	Hydrothermal	*Cornus walteri* leaves	MG	O_2_^•^^−^	e^−^-h^+^ pair separation	98.0/40	[120] 2022
				MO	•OH		97.1/50	
				MV			63.6/90	
2	CDs/TNs	Hydrothermal	Ammonium citrate (AC)	CR RhB TC	^•^OH	e^−^-h^+^ pair separation	85.9/120	[123] 2022
2	TiO_2_-MCDs	Microwave-assisted	Microalgae tablet	MB	O_2_^•−^ •OH	e^−^ trapping, Up-conversion	83/120	[121] 2022
2	CDs-BiSbO_4_	Hydrothermal	Citric acid and urea	Rh B Ciprofloxacin	O_2_^•^^−^,	Up-conversion, e^−^ trapping,	91/100	[128] 2021
					•OH,		43/100	
2	CQDs/Sb_2_WO_6_	Hydrothermal	Urea ascorbic acid	RhB	h^+^, •OH	Up-conversion, efficient e^−^/h^+^ separation	83/120	[125] 2022
2	N-CDs@ZnO composite	Hydrothermal	Malus floribunda fruits	MB	h^+^, •OH	e^−^ trapping, Up-conversion	99/60	[126] 2020
2	RCD-ZnO nanohybrid	Hydrothermal	Colocasia esculenta leaves	Dodecylbenzene sulfonate commercial detergent	h^+^, ^•^OH	e^−^/h^+^ separation	96.7/110 94.8/150	[29] 2019
2	N-CDs on BiOBr/CeO_2_	Hydrothermal	C_6_H_5_O_7_ (NH_4_)_3_ and ethylenediamine	Carbamazepine	O_2_^•−^, h^+^, •OH	Accelerating the migration and separation of the charge carries	97/120	[131] 2020
2	N,S-CQDs/TiO_2_ on polysulfone membrane	Thermal treatment	Egg yolk	Diclofenac	O_2_^•−^, •OH,	Up-conversion, efficient e^−^/h^+^ separation	62.3/150	[27] 2020
2	CQDs on BiOCl/carbonized eggshell membrane	Thermal treatment	Eggshell membranes	Tetracycline hydrochloride	O_2_^•−^, h^+^, •OH	Electron trapping	97.39/60	[25] 2021
2	ZnO/CD nanocomposites	Hydrothermal	Trisodium citrate dihydrate and ammonium carbonate	Ciprofloxacin	h^+^, O_2_^•−^, •OH	Up-conversion, efficient e^−^/h^+^ separation	98/110	[132] 2021
2	CDs modified g-C_3_N_4_/SnO_2_	Hydrothermal	Citric acid and urea	Indomethacin	O_2_^•−^, h^+^	Up-conversion, efficient e^−^/h^+^ separation	90.8/80	[133] 2021
3	CDs/MoS_2_/p-C_3_N_5_	Hydrothermal	Fungal	MB	•OH	e^−^ trapping, charge transfer	93.51/120	[124] 2022
4	Fe, N-CDs	Hydrothermal	Citric acid urea and ferric citric	MB	O_2_^•−^, h^+^, •OH	Charge separation	100/60	[120] 2022
4	CDs@P-Eu-MNs	Hydrothermal	Citric acid Cysteine	Rhodamine 6G	O_2_^•−^, h^+^	Charge separation	95/160	[122] 2022
4	C_3_N_4_-NS/CD/FeOCl	microwave-assisted	Citric acid and urea	RhB	O_2_^•−^, h^+^,	Charge separation	100/60 100/45	[127] 2021
				Tetracycline	•OH			
				hydrochloride				
4	CDs/hollow g-C_3_N_4_ nanospheres	Hydrothermal	Citric acid and urea	Naproxen	O_2_^•−^	Up-conversion, efficient e^−^/h^+^ separation	98.6/25	[134] 2020
				Indomethacin			~100/25	
				Norfloxacin			~80/25	
				Diclofenac			~50/25	
4	B-CDs on C_3_N_4_	Hydrothermal	Carbon fibers	Tetracycline hydrochloride	O_2_^•−^, h^+^	enlarged surface absorption, light-harvesting ability, and charge separation and transfer	65.82/180	[135] 2020
4	CQDs and reduced graphene oxide layers on S@g-C_3_N_4_/B@g-C_3_N_4_	Ultrasonic method	Glucose	Chloramphenicol	O_2_^•−^, •OH,	Transmission of charge	99.1/90	[136] 2020
4	(CQD) incorporated goethite (α-FeOOH) nanohybrids	Hydrothermal	Citric acid	Tetracycline	O_2_^•−^, •OH, ^1^O_2_	Up-conversion	94.5/60	[137] 2020

* Type 1, CDs; type 2, CDs/metal oxide composites; type 3, CDs/metal sulfide composites; type 4, others.

**Table 2 polymers-14-02153-t002:** Precursors, synthesis methods, quantum yields, target analytes, linear ranges, and detection limits of various CDs for the sensing applications.

Material	Precursors	Method	QY (%)	Analyte	Linear Range	LOD	Reference/Year
Si/CDs	(3-Aminopropyl) triethoxysilane and citric acid	Solvothermal	-	Hg^2+^	0–200 µM	26.7 nM	[190] 2022
CTAB/NCDs	Citric acid and urea	Solvothermal	32%	Hg^2+^	0.16–10.24 µM	85.71 nM	[171] 2022
NCDs	Citric acid and ethylenediamine	Hydrothermal	67.4%	Hg^2+^	0.3–2.0 µM	0.24 µM	[172] 2022
CDs/InPQDs@ZIF-8	Kelp powder	Hydrothermal	-	Hg^2+^	0–5 µM	8.68 nM	[173] 2022
CDs-AgNPs	Melamine and citric acid	Hydrothermal	-	Hg^2+^	100–160 µM	2.22 × 10^−8^ M	[191] 2022
NS-CDs	Aurine and citric acid	Thermal Lysis	68.94%	Hg^2+^	0–100 µM	50 nM	[192] 2022
Eu-CDs	Citric acid and urea	Hydrothermal	0.013%	Hg^2+^	0–80 µM	4 µM	[174] 2022
N-CDs/R-CDs@ZIF-8	Citric acid, urea, and spinach extract	Hydrothermal Solvothermal	-	Pb^2+^	0.05–50 µM	4.78 nM	[177] 2021
Functionalized-GQD	Graphite flakes	Ultrasonication	13.4%	Pb^2+^	0–300 µM	1.2 µM	[178] 2021
N-CDs	Sodium alginate and urea	Thermal sintering	-	Pb^2+^	-	3 ppb	[179] 2021
				Cu^2+^	-	15 ppb	
CDs-HS18	Ureibacillus thermosphaericus	Hydrothermal	17.3%	Cr^6+^	0–9 µM	36 nM	[163] 2021
N and S doped CDs	O-phenylenediamine and dl-Thioctic acid	Hydrothermal	21.82%	Cr^6+^	0–60 µM	0.64 µM	[180] 2022
CDs-Kan	Kanamycin sulfate	Hydrothermal	5.26%	Cr^6+^	0–33 μM	0.36 μM	[181] 2021
N and S doped CDs	Glycine and 4-sulfophthalic acid	Hydrothermal	-	Cr^3+^	0–40 μM	7.8 nM	[182] 2021
Orange emission CDs	1,2,4-Triaminobenzene and p-aminobenzenesulphonic acid	Hydrothermal	14.9%	Cr^3+^	1–96 μM	0.38 μM	[183] 2021
N doped CDs	Ethylene glycol and β-alanine	Heating in an oil bath	14.3%	Cr6+	0.5–500 μM	0.29 μM	[193] 2021
				4-NP	1–250 μM	0.4 μM	
N-doped CQDs	Fullerene, H_2_O_2_, and NH_4_OH	Hydroxy radical	10%	Cr^3+^	0–100 μM	2 μM	[184] 2021
CQDs	Crab-shell waste	Hydrothermal	-	Cd^2+^	50–250 µM	-	[186] 2022
N,S-CDs	Citric acid and thiourea	sonication	-	Cd^2+^	0–2.1 µM	62 nM	[187] 2021
B doped CNQDs	Urea, boric acid, and citric acid	Hydrothermal	87.4%	Cd^2+^	0–20 µM	1.1 nM	[194] 2021
				Fe^2+^	0–20 µM	2.3 nM	
N-doped CQDs	Auricularia auricular and ethylenediamine	Hydrothermal	28.4%	Cd^2+^	0–50 µM	101.55 nM	[195] 2021
				Hg^2+^	0–50 µM	77.21 nM	
CDs	1,4-Dihydroxyanthraquinone	Solvothermal	41.3%	Cu^2+^ glyphosate	50–300 ng·mL^−1^	22.65 nM 5 nM	[188] 2022
h-CDs	Hydroquinone, o-phenylenediamine, and terephthalic acid	Solvothermal	30.8%	Cu^2+^ H_2_O	0–0.01 mM	1.8 × 10^−4^ mM	[196] 2022
NCDs	Oil red O	Solvothermal	68%	Cu^2+^	0–50 μM 0–100 μM	4 nM	[189] 2021
				DMC *		50 pM	
				TC		500 pM	
				MC		5 nM	
				DC		50 nM	
				OTC		100 nM	

* Demeclocycline (DMC), tetracycline antibiotic (TC), minocycline (MC), doxycycline hydrochloride (DC), oxytetracycline (OTC).

**Table 3 polymers-14-02153-t003:** Possible applications of various CDs and their roles.

Applications	Types of CDs	Function of CDs	Reference/Year
**Bioimaging**			
Computer tomography (CT)	Barium-doped (Ba-CDs)	Contrast agents	[197] 2022
	Hafnium-doped (Hf-CDs)	Contrast agents	[198] 2020
Fluorescent imaging (FI)	Boron-doped p-phenylenediamine-based carbon quantum dots (B-PPD CDs)	Cell labeling agent	[199] 2022
	Kiwi-fruit-peel carbon dots (KFP-CDs)	Cell labeling agent	[167] 2022
Magnetic resonance imaging (MRI)	Manganese-doped blue emission carbon quantum dots (BCQD@Mn) composite	Contrast agents	[200] 2022
	Manganese-doped CDs (Mn-CDs)	Contrast agents	[201] 2021
Photoacoustic imaging (PAI)	Permeable carbon dots (PCDs)	PAI agent, Tumor ablation (laser irradiated at 1064 nm)	[202] 2022
	Carbon nitride nanoparticles (CN-NPs)	PAI agent, Tumor-growth inhibition (laser irradiated at 1064 nm)	[203] 2022
**Catalysis**			
CO_2_ reduction	N-doped carbon and carbon dots (CDs)	CO_2_ adsorbent and active N-sites to generate CH_4_/CH_3_OH by radical •CO_2_	[204] 2022
	CD-modified Co_3_O_4_/In_2_O_3_ composite	Electron and hole transfer processes	[142] 2022
Degradation of pollutants	Vis/CDs–ZIS/PS ((visible light CDs, ZnIn2S4 (ZIS), persulfate (PS))	Photoinduced charge separation	[205] 2022
H_2_ evolution	CQDs/CTF (carbon quantum dots/covalent triazine–based framework)	Up-conversion	[149] 2022
Organic synthesis	Carbon dots decorated with hydrogen sulfate groups (S-CDs)	Photocatalyst ((dehydrogenative cross-coupling (C-C bond formation) reactions))	[206] 2019
	Amine-rich N-doped carbon nanodots (NCNDs)	Photocatalyst (C-C bond formation reactions)	[207] 2019
	Citric acid–derived carbon dots (CACDs)	Photocatalyst (C-O bond photocleavage reactions)	[208] 2020
**Energy-associated application**			
Light-emitting diode (LED)	Phloroglucinol and urea precursor–based CDs, with emissions of blue (B-CDs), green (G-CDs), yellow (Y-CDs), orange (O-CDs), red (R-CDs)	Solid-state fluorescence and multicolor light emission	[209] 2022
	2,3-Diaminopyridine based CDs	Solid-state fluorescence and multicolor light emission	[210] 2022
	Gallic acid and o-phthalaldehyde–based red, green, and blue CDs	Solid-state fluorescence, multicolor light emission (CDs dispersed into epoxy resin to form multicolor LEDs)	[211] 2022
	Bio-CDs (microcrystalline cellulose and ethylenediamine)	Optical blocking films (OBF) prepared by mixing of Bio-CDs and polyvinyl alcohol (PVA) blocks the blue light	[212] 2022
Photodetectors	Nitrogen-doped graphene quantum dots (N-GQDs)	Mixing of n-type N-GQDs and SiO_2_/Si substrate to prepare the Photodetector	[213] 2022
	Pure glucose–based dual-sized CQDs	The dual-sized CQDs films directly formed on Si substrates, supporting as self-powered photodetectors.	[214] 2021
Photovoltaics	Citric acid and uric acid–based nitrogen-doped carbon quantum dots (N-CQDs)	Used as a co-sensitizer	[215] 2022
	N-CQDs (carbon and nitrogen source from Aminobenzene-dicarboxylic acid)	Hole transporter, an electron blocker	[216] 2022
Supercapacitors	CDs/NCLDH ((2D nickel–cobalt layered double hydroxide (NCLDH) nanosheets are regulated to form 3D flower-like spheres by fungus bran-derived carbon dots (CDs))	As a bridge for charge transfer	[217] 2022
	SWCNT/ZnO nanocomposite decorated with carbon dots (CDs- Citric acid, Ethylenediamine, SWCNT- single-walled carbon nanotube)	Reactive to UV light, electron-hole pairs generation	[218] 2022
Thermoelectric devices	CDs/PEDOT:PSS (poly(3,4-ethylene-dioxythiophene), poly (styrenesulphonate) nano-composite films	generation of an increased level of charge carrier concentration	[219] 2021
	PEDOT:PSS/NC@Te films ((NC- Nitrogen doped Carbon nano-dots, decorated Telluride (Te) nano-rods embedding into Poly(3,4-ethylene-dioxythiophene), Poly(styrenesulphonate))	the formation of conductive paths within the films, as well as an increase in carrier mobility and carrier concentration	[220] 2021
**Sensing**			
Colorimetric	PAA-CDs (primary aromatic amines derived CDs)	Detection of NO^2−^ ions with LOD of 0.024 μM and 0.16 μM by colorimetric and fluorimetric methods, respectively.	[221] 2022
Fluorescent	CQDs (citric acid and ethylenediamine)	Detection of Fe^3+^ and Hg^2+^ with LOD of 0.406 µM and 0.934 µM, respectively.	[222] 2022
Electrochemical	Co_3_O_4_@N-CNTs/NH_2_-GQDs/GCE composite (N-CNTs, nitrogen-doped carbon nanotubes; NH_2_-GQDs, amine-functionalized GQDs)	Detection of luteolin with a LOD of 0.1 nM	[223] 2022
	CDs/α-Fe_2_O_3_-Fe_3_O_4_ composite (CDs from 5-sulfosalicylic acid and diethylene glycol)	Detection of aflatoxin B1 With an LOD of 0.5 pM	[224] 2022
Ratiometric	dNIR-CDs (dual emission near-infrared carbon dots from glutathione and polyethylenimine)	Detection of Lysozyme with an LOD of 7 nM	[225] 2022
**Therapy**			
Antibacterial	Boron-doped glucose carbon dots (BGCDs), Sulfur-doped glucose carbon dots (SGCDs), Nitrogen-doped glucose carbon dots (NGCDs), Glucose carbon dots (GCDs)	Antibacterial activity against *Escherichia coli* and *Listeria monocytogenes*	[159] 2022
	CDs (red Korean ginseng root extract), CDs-RUT nanohybrid)	Inhibiting the growth of *Escherichia coli* (*E. coli*), *Staphylococcus aureus* (*S. aureus*)	[226] 2022
Antifungal	Nitrogen-doped glucose carbon dots (NGCDs), Sulfur-doped glucose carbon dots (SGCDs), Boron-doped glucose carbon dots (BGCDs)	Inhibiting the growth of *A. fumigatus*, *F. solani*, *P. citrinum*, *C. Albicans*, and *R. Rubra*.	[159] 2022
	Nitrogen and iodine-doped (I-CDs), i.e., I-CDs-3 (iopromide and EDA)	Inhibiting the growth of *C. Albicans*	[227] 2021
Antioxidant	Glucose carbon dots (GCDs), Nitrogen-doped Glucose carbon dots (NGCDs)	Free radical scavenging	[159] 2022
	CDs (Red Korean ginseng root extract), CDs-RUT nanohybrid	Free radical scavenging	[226] 2022
Anti-inflammatory	FCDs (fluorescent carbon dots synthesized from Carica Papaya Leaves)	Prevent red blood cells (RBC) lysis caused by induced hypotonicity.	[228] 2022
Antiviral	Hsd-CPDs (carbonized polymer dots from hesperidin (Hsd))	Hsd-CPDs surface contains bioactive moieties of apocynin, and guaiacol binds with the proteins of enterovirus A71 (EV-A71), thus blocking the viral attachment in neonatal mice.	[229] 2022
	CQDs (carbon quantum dots ethylenediamine/citric acid/boronic acid ligands)	The human coronavirus HCoV-229E is inactivated using a concentration-dependent method.	[230] 2019
Anticancer	Nano-powder of Ludox@CDs (CDs prepared from cetylpyridinium chloride)	Acts as cytotoxic to cancer cells by persuading apoptosis	[231] 2022
**Others**			
Fertilizer for Plant	nitrogen and sulfur co-doped CQDs (NS-CQDs)	Carriers of nutrients and microbes for plant growth promotion	[232] 2022
Separation of water from alcohol (alcohol dehydration)	SCQDs (sulfonated carbon quantum dots) with GO (graphene oxide)	SCQDs act as water transporter	[233] 2022
Security authentication (covertness under daylight)	NCDs printing ink (Nitrogen-doped CDs from rice straw waste)	Reversible photochromism (printed cellulose papers: no color under daylight conditions, but blue emissions under UV light)	[234] 2022
	Nano-powder of Ludox@CDs (CDs prepared from cetylpyridinium chloride)	Fingermark imaging under UV light (wide range of emission)	[231] 2022
Anti-counterfeit agent (covertness under daylight)	CDs ink (pine pollen)	Reversible photochromism (printed cellulose papers do not exhibit any color under daylight conditions, but blue emission is demonstrated under UV light)	[28] 2022
Lubricants	A-CDs (Amphiphilic CDs synthesized from TWEEN-80)	A-CDs stabilized with Span-80 are used as lubricant additives of polyalphaolefin	[235] 2022
Food packing	S-CD (Sulfur functionalized turmeric-derived carbon dots)	S-CD is used as combined material with pectin/gelatin film due to antibacterial activity against the foodborne pathogenic bacteria	[236] 2022
Flame retardant	gCDs-PET co-polyester ((gelatin based CDs as a co-polymerizable flame retardants for PET (poly(ethylene terephthalate))	Thermal decomposition of PET is catalyzed by gCDs	[237] 2022

## Data Availability

The data presented in this study are available upon request from the corresponding author.

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
