# Peer review of "Recent Advances in Synthesis, Modification, Characterization, and Applications of Carbon Dots"

_polymers, 2022, doi:10.3390/polym14112153_

Round 1

Reviewer 1 Report

Overall the review is well organized. Small improvements suggested below :

  1. In the introduction line 50-56, the sentence is too long and difficult to follow. Might be useful to break it into numbers
  2. Line 86 " This review is focused on these issues." the sentence is redundant since you say the same thing in the beginning of next paragraph
  3. Line 87-88, the word "CDs" is used too many times
  4. Either use "Fig 2" or "Figure 2" as a constant way to reference figures
  5. Fig 2 is pixelated

Author Response

  1. In the introduction line 50-56, the sentence is too long and difficult to follow. Might be useful to break it into numbers

Response:

The sentence was modified according to the reviewer’s comment.

  1. Line 86 " This review is focused on these issues." the sentence is redundant since you say the same thing in the beginning of next paragraph

Response:

The sentence was removed.

  1. Line 87-88, the word "CDs" is used too many times

Response:

The sentence was modified according to the reviewer’s comment.

  1. Either use "Fig 2" or "Figure 2" as a constant way to reference figures

Response:

“Figure” was used in the revised manuscript.

  1. Fig 2 is pixelated

Response:

Thanks for the reviewer’s comments. Figure 2, 5, 7, 8, 9, 10, 11, 12, 13, 14, and 15 were replaced by higher-quality images.

Reviewer 2 Report

Arul Pundi and Chi-Jung Chang present a review entitled "Recent advances in synthesis, modification, characterization, and applications of carbon dots".
The review is well-written and deals with a large amount of recent literature in the field of carbon (CDs) and graphene quantum dots (GQDs).
In my opinion, the review can be accepted for publication after revising some minor points listed below:

Q1) Some figures need to be of higher quality (particularly Figure 2, 7 and 8).
Q2) Section 3.2 can be briefly expanded by first introducing more fundamental and intrinsical aspects such as:
-vacancies in graphene quantum dots.
-doping and edge functionalization in graphene quantum dots.
Some interesting references, both experimental and computational, can be found in the recent literature.
Also, a previous brief description of UV-Vis absorption and emission properties of CDs and GQDs can be helpful to facilitate the reading of the 3.2.1, 3.2.2, 3.2.3 and 3.2.4 sub-sections.

Author Response

Q1) Some figures need to be of higher quality (particularly Figure 2, 7 and 8).

Response:

Thanks for the reviewer’s comments. Figure 2, 5, 7, 8, 9, 10, 11, 12, 13, 14, and 15 were replaced by higher-quality images.

Q2) Section 3.2 can be briefly expanded by first introducing more fundamental and intrinsical aspects such as:
-vacancies in graphene quantum dots.
Response:

Section 4.3

The oxygen vacancy defect results from a loss of oxygen atom from its relative position in the crystal lattice. The introduction of surface oxygen vacancy is a promising method to tune band structure, modify surface chemical states, and accelerate charge separation of photocatalysts.

-doping and edge functionalization in graphene quantum dots.
Some interesting references, both experimental and computational, can be found in the recent literature.
Response:

Section 2.3

Since doping can effectively tune the physical and chemical properties of CDs, it has recently attracted increasing attention. Doping is the introduction of a dopant, such as boron, nitrogen, chlorine, sodium, and potassium, into the structure of a CQD. Doped CDs and CDs-based composites exhibit enhanced light absorption and photoluminescence properties compared to pristine CDs. When CDs were doped with appropriate heteroatoms, their chemical composition, nanostructure, electronic structure, and catalytic properties changed because of the overlapped atomic orbitals of carbon atoms and heteroatoms, together with the electron push-pull effect of heteroatoms. [63]

Response:

Section 2.4

The properties of CDs were affected by the types of surface functional groups. The surface-modified CDs can be used for light-emitting devices, drug-releasing, chemical sensing, targeting, and extracting analytes [90]. The formation of surface functional groups such as hydroxyl, carboxyl, amine, and amide can impart higher dispersing stability of CDs in many aqueous media and solvents. It helps to improve the performance catalysis, sensing, and bio-related applications of CDs-based materials by facilitating their interaction with pollutants, analytes, and biological species. Besides, the characteristics of graphene quantum dots can be tuned by functionalizing their edge structures.

Also, a previous brief description of UV-Vis absorption and emission properties of CDs and GQDs can be helpful to facilitate the reading of the 3.2.1, 3.2.2, 3.2.3 and 3.2.4 sub-sections.

Response:

Section 3.2

The strong and tunable fluorescence of CDs have been widely investigated, because fluorescence enables the use of CDs in biomedicine, optics, catalysis, and sensing applications. Factors affecting the properties or performance of CDs are worth studying, including the size-dependent fluorescence, concentration-dependent multi-color luminescence, solvation effects, electronic structures and photophysics analysis.

Reviewer 3 Report

Here are my comments.

1. The main question is correspondence between synthesis method and possible application of carbon dots.

2. The topic is relevant in the field because it address the relationship between the method of synthesis and the properties of carbon dots especially in higher scale, as well as application connected with synthesis method.

3. The added value is analysis of correspondence between the synthesis method and properties and applicastion of carbon dots in different branches of industry.

4. There is no needed improvement of methodology or controls.

5. The conclusions are consistent with the evidence and arguments.

6. The reference are appropriate.

7. There are no additional comments regarding tables and figures.

8. According to IUPAC regulations in polymers nomenclature and properties the term "average molar mass" should be used instead of "molecular weight". The Authors should correct that term in the whole manuscript.

Author Response

  1. The main question is correspondence between synthesis method and possible application of carbon dots.
    Response: Thanks for the reviewer’s comments.

  2. The topic is relevant in the field because it address the relationship between the method of synthesis and the properties of carbon dots especially in higher scale, as well as application connected with synthesis method.
    Response: Thanks for the reviewer’s comments.

  3. The added value is analysis of correspondence between the synthesis method and properties and application of carbon dots in different branches of industry.
    Response: Thanks for the reviewer’s comments.

  4. There is no needed improvement of methodology or controls.
    Response: Thanks for the reviewer’s comments.

  5. The conclusions are consistent with the evidence and arguments.
    Response: Thanks for the reviewer’s comments.

  6. The reference are appropriate.
    Response: Thanks for the reviewer’s comments.

  7. There are no additional comments regarding tables and figures.

Response: Thanks for the reviewer’s comments.

  1. According to IUPAC regulations in polymers nomenclature and properties the term "average molar mass" should be used instead of "molecular weight". The Authors should correct that term in the whole manuscript.

Response:

It was modified according to the reviewer’s comment.